# *icepack*: a new glacier flow modeling package in Python, version 1.0

Daniel R Shapero[1], Jessica A Badgeley[2], Andrew O Hoffman[2], and Ian R Joughin[1]

[1]Polar Science Center, Applied Physics Laboratory, University of Washington, Seattle, WA, USA
[2]Department of Earth and Space Sciences, University of Washington, Seattle, WA, USA

**Correspondence:** Daniel Shapero (shapero@uw.edu)

**Abstract.** We introduce a new software package called *icepack* for modeling the flow of glaciers and ice sheets. Icepack is built on the finite element modeling library Firedrake, which uses the Unified Form Language (UFL), a domain-specific language embedded into Python for describing weak forms of partial differential equations. The diagnostic models in icepack are formulated through action principles that are specified in UFL. The components of each action functional can be substituted for different forms of the user's choosing, which makes it easy to experiment with the model physics. The action functional itself can be used to define a solver convergence criterion that is independent of the mesh and requires little tuning on the part of the user. Icepack includes the 2D shallow ice and shallow stream models. We have also defined a 3D hybrid model based on spectral semi-discretization of the Blatter-Pattyn equations. Finally, icepack includes a Gauss-Newton solver for inverse problems that runs substantially faster than the BFGS method often used in the glaciological literature. The overall design philosophy of icepack is to be as usable as possible for a wide a swath of the glaciological community, including both experts and novices in computational science.

## 1 Introduction

Nearly all glaciologists, from graduate students to senior researchers, need to use numerical models at some point in their career. Several software packages for glacier flow modeling already exist and are effective in the hands of experts. We highlight four main uses of glacier models in the literature:

1. projecting future glacier extent and estimating the sea-level rise contribution from glacier dynamics (Joughin et al., 2014),

2. exploring aspects of glacier physics, such as hydrology and calving, that are not completely understood (Nick et al., 2010; Werder et al., 2013),

3. estimating unobservable quantities, such as bed friction or rheology, from observational data (Vieli et al., 2007; Shapero et al., 2016), and

4. reconstructing what glaciers of the near- or distant-past may have looked like (Huybrechts, 2002).

To accomplish these tasks, modeling tools are usually written in compiled programming languages such as C, C++, and Fortran for reasons of computational efficiency. Many glaciologists receive little or no formal programming training, much less

in these languages, and are instead self-taught in either Python or MATLAB. The ubiquity of C, C++, and Fortran in scientific computing can create a barrier to entry for glaciologists who are not experts in high-performance computing. In this paper, we introduce a new Python software package for glacier flow modeling called *icepack*. Our goal is to make a tool that will be both accessible to novices and more productive for experts. We have focused efforts thus far on process studies of individual glaciers or drainage basins (use cases 2 and 3 of the list above). Development of icepack is ongoing and we will broaden our efforts to encompass more use cases in future.

The glacier flow modeling package closest in spirit to icepack is VarGlaS (Brinkerhoff and Johnson, 2013). VarGlaS is implemented using the finite element modeling (FEM) package FEniCS (Logg et al., 2012), whereas icepack is built on top of Firedrake, which began as an outgrowth of FEniCS. Both packages share a similar goal of saving users from manually writing low-level code for assembling the systems of equations that discretize their model physics. Instead, users describe the weak form of the partial differential equations they wish to solve using a high-level domain-specific language (DSL) called the *Unified Form Language* or UFL (Alnæs et al., 2014). This DSL is embedded entirely into Python, i.e. the complete syntax of UFL can be mapped directly onto overloaded operators in the Python programming language. Both FEniCS and Firedrake then generate optimized C or C++ code to assemble the discretized system of equations from this symbolic description of the problem (Kirby and Logg, 2006; Rathgeber et al., 2016). Combining a DSL and a code generator frees users from the very error-prone process of writing these assembly kernels themselves, and makes the syntax of the code align more closely to the syntax of written mathematical expressions. Additionally, having a high-level symbolic description of the problem makes it possible to automatically derive tangent linear models and adjoints (Mitusch et al., 2019).

Icepack improves upon the groundwork laid in VarGlaS in three main respects. First, icepack includes a simple 3D flow model that uses several features only available in Firedrake: extruded meshes and tensor product finite elements (Bercea et al., 2016; McRae et al., 2016) (see §2.2.3). Second, icepack's architecture is designed to make it easy for users to alter the various physics components, such as the rheology and basal friction, of any of its constituent models (see §2.4). Finally, the inverse solver in icepack uses the Gauss-Newton method, which converges faster and more reliably than the BFGS method used in VarGlaS (see section §4.4).

## 2 Physics

The two main components of a glacier flow model are a *diagnostic* and a *prognostic* equation. The diagnostic equation prescribes the ice velocity through a time-independent, nonlinear, elliptic partial differential equation (PDE). The prognostic equation prescribes how the ice thickness evolves through conservation of mass. Mathematically, these two coupled PDEs can be thought of as a differential-algebraic equation; see Ascher and Petzold (1998).

The evolution equations for ice thickness and velocity are necessary for any simulation, but other fields with their own dynamics can be a part of the problem as well. For example, the rheology and friction are the main coefficients in the diagnostic equation. These coefficients are functions of other fields that have their own evolution equations. The rheology is a function of temperature, englacial water content, and damage from crevassing. Likewise, the friction coefficient can be described in terms

of the subglacial water pressure and a roughness factor for the underlying bedrock (Cuffey and Paterson, 2010). The diagnostic and prognostic equations can then be supplemented with evolution equations for these fields, for example the heat equation for

temperature, or a hydrology model for subglacial water pressure.

| Symbol | Meaning |
|---|---|
| $h$ | thickness |
| $b$ | bed elevation |
| $s$ | surface elevation |
| $d$ | water depth |
| $u$ | velocity |
| $\dot{a}_s$ | surface mass balance |
| $\dot{a}_b$ | basal mass balance |
| $\dot{\varepsilon}$ | strain rate, $\frac{1}{2}(\nabla u + \nabla u^\top)$ |
| $C$ | bed friction coefficient |
| $A$ | rheology coefficient |
| $E$ | enthalpy density |
| $\nu$ | unit outward normal |
| $J$ | action functional |
| $\Omega$ | spatial domain |
| $\Gamma$ | calving terminus |

**Table 1.** Mathematical symbols

| Symbol | Meaning |
|---|---|
| $n$ | Glen's flow law exponent |
| $m$ | Weertman sliding law exponent |
| $\rho_I$ | ice density |
| $\rho_W$ | seawater density |
| $c$ | specific heat capacity of ice |
| $L$ | latent heat of melting of ice |
| $g$ | gravitational acceleration |

**Table 2.** Physical constants

In this section, we'll describe the mathematical formulation of each of the models implemented in icepack. See table 1 for definitions of all mathematical symbols and table 2 for physical constants.

## 2.1 Prognostic model

The *prognostic model* or *mass transport equation* (the two terms are synonymous) describes how the ice thickness changes in time. The prognostic model is:

$$\frac{\partial h}{\partial t} + \nabla \cdot hu = \dot{a}_s - \dot{a}_b, \tag{1}$$

where $\dot{a}_s$ and $\dot{a}_b$ are the surface and basal mass balance. This problem has the apparent form of a conservative advection equation, but the velocity $u$ is coupled to the thickness and surface slope in such a way that the whole problem is not hyperbolic. For the specific case of the shallow ice approximation (see section §2.2.1), the coupled system is parabolic. In all other cases, the problem does not have a PDE "type" in the usual sense because the velocity is found through solving an elliptic PDE where the thickness and surface slope are coefficients.

Icepack represents the thickness using continuous, piecewise polynomial basis functions in each cell of the mesh. In the examples we use up to degree 2 and the unit tests use up to degree 4. We have not yet implemented a formulation that works with discontinuous basis functions, but this extension is completely feasible within our framework.

The surface elevation is calculated as

$$s = \max\{b + h, (1 - \rho_I/\rho_W)h\}, \tag{2}$$

the first case corresponding to grounded ice and the second case corresponding to floating ice. In the immediate vicinity of the grounding line, the assumption that the floating ice is in hydrostatic equilibrium with the ocean fails. Most models assume hydrostasy and icepack does as well. Elmer/Ice, on the other hand, solves a contact problem for the moving upper and lower ice surfaces and thus can accurately model non-hydrostatic ice shelves (Gagliardini et al., 2013).

We implement two types of boundary conditions for the prognostic equation. Users can specify an inflow flux value and this value becomes a source of thickness at any point along the domain boundary where the ice velocity is pointing in to the domain. The flux at the inflow boundary can change in time. Second, we impose outflow boundary conditions on any part of the domain where the ice velocity is pointing outwards. Which segments of the boundary are inflow or outflow are diagnosed automatically by calculating the sign of the dot product between the velocity and the unit outward normal vector.

The mass transport equation for ice thickness is a free boundary problem, where the free boundary is the contour between ice-covered and ice-free regions (Schoof and Hewitt, 2013). Where there is ablation in ice-free regions, a correctly-implemented prognostic solver will compute a negative thickness, which is unphysical and requires correction. The prognostic solver in icepack truncates the thickness to zero, which approximates the free boundary, where there is ablation in ice-free regions. An implicit approach would instead treat the free boundary problem directly as a variational inequality (Jouvet and Bueler, 2012). Icepack currently lacks such a scheme and this will be the subject of future development. PETSc includes scalable solvers for variational inequalities (Bueler, 2020) that are also available through Firedrake.

## 2.2 Diagnostic model

There are four *diagnostic* or *stress balance* models implemented in icepack. For each of the diagnostic models, we use a formulation of the physics based on a *minimization* or *action principle* (Dukowicz et al., 2010). Action principles are completely equivalent to the usual weak form of a partial differential equation but have certain numerical advantages, as described below.

The most complex and most physically accurate model that we implement is the *first-order* or *Blatter-Pattyn* approximation, a 3D system describing the horizontal velocity (Blatter, 1995; Pattyn, 2003). (In icepack we refer to our implementation of this equation as a *hybrid model* as it includes both shear and plug flow modes, described below.) The first-order model is based on an asymptotic expansion of the Stokes equations with respect to the ratio of the ice thickness to a typical horizontal length scale. The aspect ratio of most glacier flows is on the order of 1/20 or less, although there are some exceptions. For example, the main trunk of Jakobshavn Isbrae in Greenland flows through a very deep and narrow trough with an aspect ratio closer to 1/5.

From the first-order model, two approximations are possible. First, the *shallow ice approximation* (SIA) comes from assuming that vertical shear is the dominant mode of ice flow. The SIA model (Hutter, 1981) describes the interior of ice sheets well, where flow can be simply described as bed-parallel shear and surface and basal slopes are small. This approximation breaks down near ice margins, where basal sliding can be a large fraction of the surface speed and where membrane stresses are substantial. Second, one could assume that ice flow is purely by horizontal extension and that the surface and bed velocities are practically the same, which is called the *shallow stream approximation* (SSA). The SSA model describes the fast-flowing margins of the ice sheet best, encompassing grounded ice streams or floating ice shelves (MacAyeal, 1989).

All the diagnostic models inherit a fundamental nonlinearity in the mechanics of ice flow. For a Newtonian viscous fluid, the stress tensor $\tau$ and the strain rate tensor $\dot{\varepsilon}$ are linearly proportional to each other. Glaciers, however, have a nonlinear constitutive relation:

$$\dot{\varepsilon} = A|\tau|^{n-1}\tau \tag{3}$$

where $A$ is the *fluidity* and $n$ is the *Glen flow law exponent* (Cuffey and Paterson, 2010). The most commonly-used value of the flow law exponent is $n \approx 3$ as determined from laboratory experiments. Strictly speaking, the fluidity should be a tensor field, but is almost universally treated as a scalar (Gillet-Chaulet et al., 2006).

All of the diagnostic models in icepack are described through *variational* or *action* principles (Dukowicz et al., 2010). Rather than describe the velocity as the weak solution of a nonlinear PDE, an action principle instead states that the velocity minimizes a functional called the action. The action consists of four terms:

$$\text{action} = \iint \text{stress} \times \text{strain rate} \, dz \, dx - \int \text{basal friction} \times \text{sliding velocity} \, dx$$
$$- \iint \text{surface slope} \times \text{velocity} \, dz \, dx - \iint \text{ocean pressure} \times \text{velocity} \, dz \, d\gamma \tag{4}$$

where $dz \, dx$ denotes integration over the entire glacier, $dx$ denotes integration over the glacier footprint, and $dz \, d\gamma$ over the side wall boundary. The action has units of power (energy/time) and can be related to the rate of decrease of the thermodynamic free energy (De Groot and Mazur, 2013).

Every diagnostic model in icepack is encapsulated in its own Python class. The key responsibility of the model classes is to take in the input fields – the ice thickness, velocity, etc. – and return a symbolic description of the action functional in UFL.

The action principle is especially useful for designing a robust numerical solver, the implementation of which will be described in §4.2. For viscous flow problems near to steady-state, the action is *convex* as a function of the ice velocity, i.e. its second derivative is positive-definite. Convexity implies that the action functional has a unique minimizer and that, with an appropriate line search strategy, Newton's method will converge from any initial guess. Algorithms for minimizing convex functionals have better convergence guarantees than algorithms for solving general nonlinear systems of equations while having no additional computational cost (Nocedal and Wright, 2006). Both formulations are mathematically equivalent.

### 2.2.1 Shallow ice approximation

The shallow ice model can be derived from the Blatter-Pattyn approximation by assuming ice flow is dominated by bed-parallel shear and that surface and basal slopes are small, i.e. $\partial u/\partial x \ll \partial u/\partial z$. The result is a 2D system of equations for depth-averaged horizontal velocity. The class in icepack that represents this physics model is called `ShallowIce`. The terms in the action functional are:

$$\text{localization} = \frac{1}{2} \int_\Omega u \cdot u \, dx \tag{5}$$

$$\text{gravity} = \int_\Omega \frac{2A(\rho_I g)^n}{n+2} h^{n+1} |\nabla s|^{n-1} \nabla s \cdot u \, dx \tag{6}$$

$$\text{penalty} = \frac{1}{2} \int_\Omega \ell^2 \nabla u \cdot \nabla u \, dx \tag{7}$$

and the action functional is

$$J = \text{localization} + \text{gravity} + \text{penalty}. \tag{8}$$

The default value of the length scale $\ell$ in the penalty term is defined as:

$$\ell = 2 \max\{\text{cell diameter}, 5h\} \tag{9}$$

but users can adjust this to the value of their choice. This penalty term smooths over numerical artifacts, especially near the ice margins and termini. In these regions the shallow ice approximation is less applicable, so the error in solving a different set of equations is small compared to the inherent modeling error in using these equations in the first place.

We verified the correctness of our implementation by checking that numerical results converge at the expected rate to the analytical Bueler profile for a circular symmetric ice sheet (Greve and Blatter, 2009).

The shallow ice approximation applies well in ice-sheet interiors. For this reason, and because the equations are particularly simple to solve, this approximation has been a common choice for ice-sheet modeling (Cuffey and Paterson, 2010; Kirchner et al., 2016). This approximation does not work well in areas of the ice sheet where the flow has substantial membrane stresses, like fast outlet glaciers.

### 2.2.2 Shallow stream and shelf approximations

The shallow stream model can be derived from the Blatter-Pattyn approximation by assuming nearly plug flow, i.e. $\partial u/\partial z \ll \partial u/\partial x$. The momentum equations can, again, be vertically integrated to obtain a 2D system of equations. The class in icepack that represents this physics model is called `IceStream`. The terms in the action functional are:

$$\text{viscosity} = \frac{n}{n+1} \int_\Omega hA^{-\frac{1}{n}} |\dot\varepsilon(u)|^{\frac{1}{n}+1} dx \tag{10}$$

$$\text{friction} = \frac{m}{m+1} \int_\Omega C|u|^{\frac{1}{m}+1} dx \tag{11}$$

$$\text{gravity} = - \int_\Omega \rho_I gh\nabla s \cdot u \, dx \tag{12}$$

$$\text{terminus} = \frac{1}{2} \int_\Gamma (\rho_I gh^2 - \rho_W gd^2) u \cdot \nu \, d\gamma \tag{13}$$

and the action functional is

$$J = \text{viscosity} + \text{friction} - \text{gravity} - \text{terminus}. \tag{14}$$

When the ice is floating, there are two simplifications: the friction coefficient $C$ is 0, and the surface elevation $s$ can be written in terms of the thickness $h$ as

$$s = (1 - \rho_I/\rho_W)h. \tag{15}$$

Additionally, the terminal stress term of the action disappears after applying integration by parts to the gravity term to shift the gradient of the surface elevation over onto the velocity. Since the action functional for ice shelves has fewer terms than for grounded ice streams, we have defined a separate `IceShelf` model class. The ice shelf and ice stream models share common components, i.e. the viscosity and side wall stress.

We verified the correctness of our implementation of the ice shelf model by checking that numerical results converge at the expected rate to an analytical solution for the velocity with a linearly sloping thickness in a rectangular domain (Greve and Blatter, 2009). There are analytical solutions to the shallow stream equations with basal friction (Böðvarsson, 1955; Bueler, 2014); these solutions are more complex algebraically than those for the case without basal friction. To get an exact solution, we chose the ice velocity and thickness a priori and used the computer algebra system SymPy (Meurer et al., 2017) to generate a friction coefficient that makes these fields an exact solution. We hand-tuned the input parameters so that the values of the manufactured friction coefficient were within reasonable bounds. We then checked that numerical results converge to this manufactured solution at the expected rate (Roache, 2002).

Several studies have compared the shallow stream approximation to 3D models such as Blatter-Pattyn or full Stokes (Pattyn et al., 2013). The most appreciable difference between lower- and higher-order models occurs near the glacier grounding line, where the full Stokes equations can represent bridging stresses (Van der Veen, 2013). The lack of vertical strain rates in the

ice viscosity in 2D models can also lead to different equilibrium grounding line positions under the same external forcing. The advantage of the SSA is that that it can capture most of the overall flow features of fast-flowing glaciers but at much lower computational cost than the Stokes equations.

### 2.2.3 Hybrid model

The first-order model in icepack is described in the class `HybridModel`. The shallow ice and shallow stream models follow directly from the variational principles described above. The hybrid flow model, while also based on a variational principle, uses two more advanced mathematical techniques: terrain-following coordinates and spectral discretization in the vertical dimension.

Each of these techniques has appeared in the literature on glacier modeling before but rarely all in the same place for a 3D model. Langdon and Raymond (1978) and Bassis (2010) used variational principles and vertical spectral methods, but these works considered only flowband models. Kleiner and Humbert (2014) used terrain-following coordinates for 3D glacier flow modeling, but they discretized the problem with finite difference methods in every direction and did not take into account the variational formulation of the diagnostic model. Jouvet (2015) used vertical semi-discretization of the variational problem, but this model used a finite difference discretization in the vertical direction. The model used in Brinkerhoff and Johnson (2015) is the closest to the one we present below. This work used both terrain-following coordinates and a tensor product basis of Lagrange finite elements in the horizontal and higher-degree polynomials in a single vertical layer. For the vertical basis functions, they used a plug flow mode and one shear mode.

**Terrain-following coordinates**. Rather than the usual Cartesian coordinate system, the hybrid flow model uses terrain-following coordinates. The terrain-following vertical coordinate $\zeta$ is

$$\zeta = \frac{z - b}{h} \tag{16}$$

where $b$ is the ice base. We can then think of the computational domain as the Cartesian product of a 2D footprint domain $\Omega$ and the unit interval $[0, 1]$.

Both the bed elevation and thickness depend on $x$ and $y$. As a result, the formula for the horizontal gradient of a field in terrain-following coordinates includes an additional geometric correction factor. Letting $\nabla_z$ and $\nabla_\zeta$ denote the horizontal gradient with respectively $z$ and $\zeta$ held constant, the chain rule gives us that

$$\nabla_z q = \nabla_\zeta q + \frac{\partial q}{\partial \zeta} \nabla \zeta, \tag{17}$$

where we can calculate the spatial gradient of $\zeta$ as

$$\nabla \zeta = -h^{-1} \left\{ (1 - \zeta) \nabla b + \zeta \nabla s \right\}. \tag{18}$$

Likewise, the strain rate of a vector field can be expressed as

$$\dot{\varepsilon}_z(u) = \dot{\varepsilon}_\zeta(u) + \frac{1}{2} \left( u \otimes \nabla \zeta + \nabla \zeta \otimes u \right) \tag{19}$$

where $\otimes$ is the tensor product of two vectors.

For the Stokes equations, this alternative coordinate system also helps avoid the problem of how to enforce the condition $u \cdot \nu = -\dot{a}_b$ at the ice base, where $\nu$ is the unit outward normal vector and $\dot{a}_b$ is the basal mass balance. This boundary condition is difficult to impose exactly because the unit outward normal vector $\nu$ is defined on mesh faces while the velocity is defined at mesh vertices. Elmer/Ice uses an ad-hoc procedure to define the unit normal vectors at mesh nodes (Gagliardini et al., 2013). This procedure is nearly always effective in practice. But with a transformation to terrain-following coordinates, we can set the terrain-following vertical velocity $\omega$ to be $-\dot{a}_b/h$ at the ice base to impose this boundary condition with no additional intervention.

We also argue that answers expressed in terrain-following coordinates are more intuitive in some respects than in Cartesian coordinates. At the bed of a grounded glacier, the vertical velocity in Cartesian coordinates is

$$w = -\dot{a}_b + u \cdot \nabla b. \tag{20}$$

Knowing that a model gives a vertical velocity at the base of a glacier of, say, 10 cm/year, the modeler needs to also know the bed slope and sliding velocity. In other words, it is not immediately clear whether the vertical velocity is a result of basal mass balance or of geometry without additional information. By contrast, the vertical velocity $\omega$ in terrain-following coordinates evaluated at $\zeta = 0$ is completely determined by basal mass balance and ice thickness.

**Spectral discretization**. Terrain-following coordinates open up several choices for how to describe the vertical variation of the velocity field. In Elmer/Ice, for example, the user can extend the finite element discretization into a number of vertical layers. The number of vertical layers is a user-tuneable parameter, depending on the desired resolution along this axis (Gagliardini et al., 2013).

The horizontal velocity for many realistic flows is very smooth as a function of depth and this suggests a different approach. For example, under the plug flow approximation, the horizontal velocity is constant with depth. Under the shallow ice approximation, the horizontal velocity varies with depth as $1 - (1 - \zeta)^{n+1}$ where $n = 3$ is the Glen flow law exponent. This extra information about our solution suggests a modal rather than a nodal discretization strategy.

Rather than divide the spatial domain into many vertical layers, we can instead use only one vertical layer and increase the polynomial degree in the vertical direction to obtain higher resolution. This type of basis, in which different shape functions are used in different dimensions, is called a *tensor product* element (McRae et al., 2016). Given a set of finite element basis functions $\{\phi_k(x,y)\}$ defined on the 2D domain $\Omega$ and a set of basis functions $\{\psi_l(\zeta)\}$ defined on the unit interval $[0,1]$, the tensor product finite element basis $\{\Phi_{kl}\}$ on the extruded domain is defined as

$$\Phi_{kl}(x,y,\zeta) = \phi_k(x,y)\psi_l(\zeta). \tag{21}$$

For example, we can use piecewise linear or quadratic elements on (horizontal) triangles and use quintic or higher degree polynomials in the vertical. Rather than use the usual Lagrange elements in the vertical dimension, we can instead use the Legendre polynomial basis. The Legendre polynomials are mutually orthogonal, which makes the mass matrix block-diagonal, and provide better approximation properties (Szabó et al., 2004). Our main reason for choosing to build icepack using the Firedrake package was because it natively supports tensor product elements.

The combination of using extruded meshes and tensor product elements in the vertical direction can be thought of merely as a way to discretize a PDE that has special structure. Alternatively, we can view discretization in the vertical as defining a family of models indexed by the number of vertical basis functions. The order-$d$ model defines a coupled system of PDEs for $d$ vector fields. Each vector field represents one mode of vertical variability, similar to the distinction between barotropic and baroclinic modes in atmospheric physics and oceanography. The system is then discretized in the horizontal and solved numerically. In any case, the code is the same regardless of how one views the underlying mathematics.

The user then has to decide how many vertical modes are sufficient. Using only degree 0 is exactly equivalent to the shallow stream approximation and we use this fact as a sanity test for the hybrid model. The degree-2 model is the minimal model that still exhibits vertical shear and can satisfy the zero-stress boundary condition at the ice surface. Going up to a degree-4 model is sufficient to capture the exact solution for the shallow ice approximation. In the tutorial notebooks for icepack, we use up to degrees 2 and 4, but the test suite checks up to degree 8.

Brinkerhoff and Johnson (2015) used a similar approach to the one described above, with one vertical basis function for plug flow and one for shear flow. The shear flow basis function was chosen to be exact assuming the SIA balance with cold ice, together with a heuristic approximation for polythermal ice. By contrast, our approach allows for an arbitrary number of shear modes.

**Action functional.** We can now describe all of the terms in the action functional for this model:

$$\text{viscosity} = \frac{n}{n+1} \int_\Omega \int_0^1 h A^{-\frac{1}{n}} \sqrt{|\dot\varepsilon(u)|^2 + h^{-2}|\partial_\zeta u|^2}^{\frac{1}{n}+1} \, d\zeta \, dx \tag{22}$$

$$\text{friction} = \frac{m}{m+1} \int_\Omega C|u(\zeta=0)|^{\frac{1}{m}+1} dx \tag{23}$$

$$\text{gravity} = - \int_\Omega \int_0^1 \rho_I g h \nabla s \cdot u \, d\zeta \, dx \tag{24}$$

$$\text{terminus} = \int_\Gamma \int_0^1 (p_I - p_W) u \cdot \nu \, d\zeta \, d\gamma \tag{25}$$

where $p_I$ is the ice overburden pressure and $p_W$ is the water pressure (if any) at the calving terminus from the ocean or a pro-glacial lake. The full action functional is exactly analogous to that of the shallow stream model:

$$J = \text{viscosity} + \text{friction} - \text{gravity} - \text{terminus} \tag{26}$$

Again, we note that the expression for horizontal the strain rate in terrain-following coordinates comes from equation (19), which includes the additional geometric correction factor.

In terrain-following coordinates, the ice overburden pressure at relative depth $\zeta$ is $p_I = \rho_I g h(1-\zeta)$. The water pressure is zero above the waterline and increases linearly below it. Letting $\zeta_{\text{sl}}$ denote the relative depth to the waterline, the water pressure is

$$p_W = \rho_W g h(\zeta_{\text{sl}} - \zeta)_+ \tag{27}$$

where $w_+$ denotes the positive part of a real number. This quantity is continuous but only piecewise linear. Integrating it correctly requires more work above and beyond the usual symbolic approach; see section §4.5.2 for details.

## 2.3 Boundary conditions

The shallow shelf, shallow stream, and hybrid models require boundary conditions to be well-posed. We implement three different types which are largely the same for each physics model.

1. Users can specify the velocity on segments of the domain boundary where ice is flowing in; this is a Dirichlet condition which is eliminated from the system at the level of the solver. Supplying inflow boundary conditions for the hybrid model necessarily requires more information, namely the variation of the horizontal velocity with depth.

2. The Neumann condition at the ice terminus (equation (13)) is a natural boundary condition which is imposed by adding the relevant term to the action functional.

3. We allow for side wall boundary conditions where the ice has non-zero thickness but abuts, say, a fjord wall or an ice shelf embayment that exerts resistive stresses. In the normal direction, the boundary condition is that $u \cdot \nu = 0$ where $\nu$ is the unit outward normal vector. In the tangential direction, the boundary condition is $hM \cdot \nu = -C|u|^{\frac{1}{m}-1}u$ where $C$ is a side wall friction coefficient and $M$ is the membrane stress tensor. Putting these together gives a mixed Dirichlet-Robin boundary condition. The side wall friction (Robin) boundary condition is natural and thus can be added directly to the action functional. The constraint of no normal flow is harder to impose directly for a curved boundary; we used the penalty method.

Future versions of the code will use Nitsche's method (Nitsche, 1971) instead of the penalty method to enforce the no normal flow constraint. Nitsche's method dramatically improves the conditioning of the associated optimization problem. Our implementation of the hybrid model requires special care to handle the terminus boundary condition as a consequence of our choice of vertical basis functions; see section §4.5.2.

The shallow ice model alone is able to cope with ice-free regions in the domain, albeit with truncation at 0 thickness. All of the remaining models and their boundary conditions assume that the entire spatial domain is ice-covered. We will improve the solvers to address ice-free areas in future versions.

## 2.4 Substituting model components

Many aspects of glacier physics are not completely understood. For example, the most commonly used sliding law is the power law

$$\tau_b = -C|u|^{\frac{1}{m}-1}u \tag{28}$$

for some exponent $m$. Older research assumed $m = 3$ based on the theory of regelation (Weertman, 1957), which has since been referred to as the Weertman sliding law. When $m = \infty$, the basal shear stress is independing of the sliding speed; this

is referred to as perfect plasticity. More recently, Schoof (2005) proposed an alternative form that acts like the Weertman law when $|u| \ll u_c$ and like the plastic law when $|u| \gg u_c$. A simplified form of this model can be expressed as

$$\tau_b = -C \left( \frac{|u|}{|u| + u_c} \right)^{\frac{1}{m}} \frac{u}{|u|} \tag{29}$$

where $u_c$ is some critical speed. This latter equation has been found to agree best with laboratory experiments on till (Zoet and Iverson, 2020) and in reproducing observed velocity variations (Joughin et al., 2019).

The Weertman and plastic sliding laws possess the same functional form but differ only in the value of a single scalar parameter $m$. The Schoof sliding law, on the other hand, has a totally different functional dependence on the velocity. Several authors, including Schoof, have proposed that the basal shear stress is also a function of the effective pressure $N = \rho g h - p_w$ within the subglacial hydrological system (Budd et al., 1979; Schoof, 2005). Implementing these more sophisticated mathematical models would require adding an extra argument to the procedure for solving the diagnostic equation.

One of our goals with icepack is to facilitate experimentation with the model physics, even for novice users. Of the programming languages that are commonly used for scientific computing, only Python and possibly Julia would appear to meet these needs. To support use cases like implementing the Schoof sliding law, it must be possible not just to change the value of a single parameter but to completely alter the functional form of a given model component. For uses cases like explicitly adding the dependence of basal shear stress on hydrology, it must also be possible to add entirely new fields to a given model component. In programming terms, this amounts to changing the number of arguments to the function that calculates basal shear stress, which Python accomodates easily. For a library developed in C or Fortran, the user would then also have to change the signature of the diagnostic solve function, which is undesirable. In C++, one could avoid changing the signature of the diagnostic solve routine by (1) using variadic templates, (2) wrapping the inputs in a class, or (3) passing all arguments in a dictionary. Using variadic templates or wrapping the inputs in a class would require users to know more about generic or object-oriented programming than a novice might. Using a dictionary data structure to pass arguments is relatively easier but would be more idiomatic in Python than in C++. Some of the difficulty associated with changing the model physics in a C++ code can be alleviated with automatic differentiation tools, which have been used successfully in Albany/FELIX (Tezaur et al., 2015) and ISSM (Hück et al., 2018). Nonetheless, C++ or any language with a stronger type discipline will be much more restrictive about changing function signatures.

In icepack, users can substitute any diagnostic model component for the parameterization of their choosing, including adding new fields. From the user's perspective, substituting model physics components does not require any advanced language features beyond keyword arguments. To understand how this is possible, we will briefly describe the path that the input fields take through the program.

1. The user passes all arguments to the diagnostic solve procedure by keyword.

2. The diagnostic solve procedure creates a symbolic representation of the action functional by summing up several terms, like the viscosity, basal friction, etc. Calculating these terms is delegated to specialized procedures for each term. Each term procedure gets the entire collection of fields.

3. The routine that calculates the terms of the action selects which fields it actually needs from the argument dictionary, and then creates the symbolic representation of that term.

4. Finally, once the symbolic representation of the action functional has been created, all the responsibility passes to the nonlinear solver.

To substitute different model components, the user intervenes at step 3. Each model object – shallow ice, shallow stream, etc. – is initialized with a default set of routines to calculate the terms of that model's action functional. The user can replace these default routines with one of their own choosing by passing the function of their choice when that model object is initialized.

In step 3, any fields that were unnecessary for calculating a given component are ignored. For example, the gravitational driving power routine will pull out the velocity, thickness, and surface elevation. While the routine will not use them, it also has access to other fields, for example the ice fluidity.

In adopting this approach, we are restricted to using keyword arguments instead of positional arguments. We argue that employing only keyword arguments is a strength rather than a weakness because it enhances readability and comprehensibility for the particular use case of calling a physics solver. The user only needs to know the argument names, which are chosen to agree with the English name most commonly used in the literature – "friction", "rheology", "velocity", etc. The order of the arguments is arbitrary and immaterial. The preference for argument passing by name is specific to this use case, however, and is not universal.

## 2.5  Heat transport

We implemented the enthalpy transport model described in Aschwanden et al. (2012). The enthalpy density $E$ describes the heat content of the material in a way that incorporates both temperature $T$ and latent heat stored in meltwater:

$$E = \rho_I(cT + Lf) \tag{30}$$

where $L$ is the latent heat of melting of ice and $f$ is the melt fraction. The melt fraction can only be positive when the ice as at the pressure-melting point. The temperature and meltwater fraction can be uniquely calculated at any point from the value of the enthalpy, so nothing is sacrificed in describing heat content one way or the other. Using the enthalpy has the advantage of circumventing many of the difficulties associated with tracking the interface between cold and temperate ice. Our implementation of the model uses all of the simplifying assumptions described in Aschwanden et al. (2012), for example that horizontal diffusion is negligible and that heat capacity and conductivity are not temperature-dependent within each phase. The resulting PDE is

$$\left(\frac{\partial}{\partial t} + \nabla \cdot u\right)E - \frac{\partial}{\partial z}\alpha\frac{\partial E}{\partial z} = q \tag{31}$$

where $u$ is the full 3D velocity, $\alpha$ the (temperature-dependent) thermal diffusivity, and $q$ the volumetric heat sources. This form assumes some amount of vertical diffusion of water at the melting point. A realistic treatment would require treating the ice as a porous medium and modeling englacial water transport explicitly, which we have not done. The boundary condition at the

ice base is a fixed flux of heat from geothermal sources or from the oceans. We depart from Aschwanden et al. (2012) in one respect. Rather than fix the ice surface temperature to the atmospheric temperature, we instead use a Robin boundary condition that makes the ice temperature adjust to the external temperature:

$$-\alpha \frac{\partial E}{\partial z}\bigg|_{z=s} = \kappa(E - E_s) \qquad (32)$$

where $\kappa$ is a surface exchange coefficient and $E_s$ is the temperature-equivalent enthalpy that forces the surface. Using a Robin

instead of a Dirichlet boundary condition at the surface allows for more gradual adjustment to above-freezing temperatures that might otherwise create areas of unrealistically high melt fraction. Except in blue ice areas, the thermal contact between ice and the atmosphere is mediated by firn, which we do not model explicitly. The Robin boundary condition is a compromise to work around this missing component of the system. Finally, if there is an upstream boundary the user must provide the inflow values of the enthalpy with depth, while outflow boundary conditions are applied at the glacier terminus.

The heat transport model assumes that the user will supply a volumetric heating rate, but the model itself does not calculate the heating rate from other fields. The simplest description for the shear heating rate is

$$q = \tau : \dot\varepsilon = A^{\frac{-1}{n}} |\dot\varepsilon(u)|^{\frac{1}{n}+1}. \qquad (33)$$

The fluidity factor $A$ in Glen's law is roughly a known function of both temperature and melt fraction and we have included this function in the package. Parameterizing fluidity is not sufficient by itself however. While the dependence of fluidity on

temperature is known fairly well from laboratory experiments, the dependence on melt fraction is known with much less certainty (Cuffey and Paterson, 2010). Some users of this module may want to substitute in their own parameterization for melt fraction dependence. Other processes such as damage, fabric development, and impurities can affect the fluidity as well. Moreover, there may be other volumetric heat sources that users wish to account for, such as cryo-hydrologic warming (Phillips et al., 2010). These considerations defy any attempt to have one function that calculates the volumetric heating rate from the

other state variables. Instead, we opted for an interface where users calculate this rate themselves, with tutorials and examples that show how to do so for simple scenarios, leaving the freedom to alter the thermal feedbacks as they see fit. Our general design principle is that icepack will solve differential equations for the various prognostic fields but it's up to users to decide how these fields are coupled.

     We have not implemented a model for the surface or englacial transport of meltwater, although it would be possible to

include this process. Instead, we rely on an external scheme to act as a sink of enthalpy for meltwater fraction values above some user-defined critical value. A common choice for this critical value is 1% (Aschwanden et al., 2012).

## 2.6   Damage transport

Other physical fields besides temperature can influence ice fluidity. At large spatial scales (> 5 km), crevasse fields affect ice flow by reducing the lateral area over which stress can be transmitted. Modeling individual fractures is not computation-

ally feasible for large-scale simulations. Instead, we have implemented the phenomenological model described in Albrecht and Levermann (2014), which is based on the theory of continuum damage mechanics. This model is defined in the class

`DamageTransport`. Prognostic damage models can be broken down into three parts: (1) evolve the damage field based on the membrane stress of the glacier, (2) advect the damage field with flow, and (3) feed the damage field back into the fluidity of the glacier. Changes in fluidity in turn affect the membrane stress; the coupling between bulk damage and ice flow goes both ways. The model from Albrecht and Levermann (2014) adds sources of damage where the membrane stress exceeds a critical value and sinks of damage when the principal strain rate is less than a critical value.

## 3  Data assimilation

Icepack includes a set of routines for estimating the basal friction or rheology coefficients from observational data. Mathematically, this inverse problem amounts to finding a critical point of the functional

$$L(u, \theta, \lambda) = J(u) + R(\theta) + \langle F(u, \theta), \lambda \rangle \tag{34}$$

where $J$ is the misfit between the computed velocity $u$ and observations, $\theta$ is the field to be inferred and $R$ a regularization functional that measures the spatial variability of this field, $F$ is the diagnostic physics, and $\lambda$ a Lagrange multiplier to enforce the physics (MacAyeal, 1992; Joughin et al., 2004; Larour et al., 2005; Shapero et al., 2016). The optimal value $\theta$ gives the best fit to observations subject to the constraints of the physics and that it should not overfit to noise in the data. The class `InverseProblem` represents the specification of an inverse problem, which requires:

- the model object and the method that solves the diagnostic equation,

- the objective and regularization functionals,

- the observed field and the name of the argument to the diagnostic solver,

- an initial guess for the field to be estimated and the name of the argument to the diagnostic solver, and

- extra data passed to the diagnostic solver such as boundary conditions.

This class currently assumes that the observed state is always the ice velocity. The state to be estimated can be any single input field to the diagnostic model – basal friction, rheology, or another field that the user has added by customizing the model. In principle the same design would suffice for more complicated inverse problems and this is an area of active development.

The inverse problem class is flexible enough to account for users defining their own parameterization for the rheology or friction coefficient as described in §2.4. This flexibility with respect to parameterization is not just convenient but essential for common data assimilation workflows. Nearly all studies in the literature introduce a parameterization of the field to be estimated in terms of some auxiliary field in order to guarantee positivity (MacAyeal, 1992; Joughin et al., 2009). For example, one could define the friction coefficient $C$ in terms of an auxiliary variable $\beta$ as $C = \beta^2$ to guarantee positivity. One could also just as easily use $C \propto \exp(\beta)$. The data assimilation routines in icepack can work the same way for any parameterization of the physics because Firedrake provides a rich set of routines for symbolically calculating functional derivatives. The inverse problem class only needs to know which functional needs to be differentiated with respect to which field.

The `InverseSolver` class is responsible for actually solving the inverse problem. This class will be described further in section §4.4.

## 4 Numerics

In the previous sections we described *what* problems icepack can solve, i.e. various physics models and data assimilation problems. In this section, we'll describe *how* these problems are solved. This separation between the two questions parallels the broader design of the software package.

The key classes that users interact with are flow models and solvers. The role of the model classes is to describe what problem is being solved. These classes describe the diagnostic model by taking in the input fields – ice velocity, thickness, 440 surface elevation, etc. – and returning a symbolic representation of the action functional. There are several model classes, one for each set of physics equations: `ShallowIce`, `IceShelf`, `IceStream`, `HybridModel`. The model classes do not dictate how that problem should be solved numerically; this is the realm of a separate class called `FlowSolver`. This flow solver class has methods for computing the solutions of the diagnostic and prognostic equations and works the same regardless of which model is being solved. The diagnostic solve method amounts to invoking an external Newton solve procedure on the 445 symbolic action functional that the model object creates. The Newton solver itself is completely standard but the convergence criterion is not (see section §4.3). Finally, the flow solver has a method to update the ice thickness from the current value, the ice velocity, and the mass balance rates.

The Unified Form Language for specifying weak forms of PDEs contains all of the primitives necessary to express individual terms of the action functional. These primitives consist of the basic vector calculus operators like the gradient of a field, tensor 450 calculus operations like taking the dot product of two vectors or tensors, scalar functions like the square root or exponential, and symbolic integration over the mesh or its boundary. For example, the strain rate for a given velocity field $u$ can be written as `sym(grad(u))`, where the function `grad` represents the symbolic gradient of a field and `sym` represents the symmetrization of a rank-2 tensor.

### 4.1 Advective transport

Icepack offers two timestepping schemes for solving the prognostic model (equation (1)). For testing purposes we include the implicit Euler scheme, which is first-order accurate and unconditionally stable for the advection equation. The default is a second-order accurate scheme based on an implicit version of the Lax-Wendroff method, which is also unconditionally stable for the advection equation (Donea and Huerta, 2003). Explicit schemes all require a timestep that satisfies the Courant-Friedrichs-Lewy (CFL) stability condition, which many glaciologists may be unfamiliar with. With an unconditionally stable 460 scheme, users will get an answer, rather than a runtime error, should they try to use a larger timestep. The extra computational cost of using an implicit time discretization for the prognostic equation is dwarfed by the cost of the diagnostic solve. Advanced users who are interested in maximizing performance can subclass the solver to implement a faster explicit scheme. We note

that, while we the stability properties of different schemes for the linear advection equation guided our choice of method, the coupled system for both thickness and velocity is not linear and not hyperbolic.

The implicit Euler and Lax-Wendroff schemes tend to diffuse out sharp discontinuities that may be present in the true solution (Donea and Huerta, 2003). Since the ice thickness does not possess shockwaves or propagating discontinuities this error mode is tolerable. The coupling of ice thickness to velocity makes the whole system more resemble a parabolic problem than a hyperbolic one, and under the shallow ice approximation the system is truly parabolic.

Other problems in glaciology have more of a hyperbolic character. For example, the thresholding behavior of the source
terms for the damage model (see §2.6) can create sharp discontinuities. The implicit Euler scheme would obscure this important feature. For the damage solver, we have instead used a strong stability-preserving Runge-Kutta method in time, a discontinuous Galerkin basis in space, and a flux limiter to best capture these sharp interfaces (Shu and Osher, 1988).

Icepack currently lacks an adaptive timestepping scheme. Implicit schemes allow taking longer timesteps than explicit ones, but taking very long timesteps will give inaccurate solutions and, in the presence of ablation, may yield negative thickness
values. At present, users are still responsible for checking the accuracy of their results, for example by running at more than one resolution. Adaptive timestepping will be added in a future release.

## 4.2    Convex optimization

The action functional for each diagnostic model is convex, i.e. the second derivative is strictly positive-definite. From a theoretical persepctive, convexity guarantees that the problem has a unique solution. This property is also especially advantageous
for implementing numerical solvers. We use a damped Newton method to solve the diagnostic equations. Starting from a guess $u_k$ for the velocity, the *search direction* $v_k$ is the unique solution of the linear system

$$d^2 J(u_k) \cdot v_k = -dJ(u_k). \tag{35}$$

The next approximation for the velocity minimizes $J$ along the search direction $v_k$ starting from $u_k$, i.e.

$$u_{k+1} = u_k + \alpha_k \cdot v_k,$$
$$\alpha_k = \text{argmin}_\alpha J(u_k + \alpha v_k). \tag{36}$$

For an initial guess sufficiently close to the exact solution, the undamped Newton method ($\alpha_k = 1$ at every step) converges quadratically. The line search step ensures that the method can converge even from a poor initial guess, provided that the line search method satisfies the Armijo-Wolfe criteria (Nocedal and Wright, 2006).

For a convex problem, $d^2 J$ is a symmetric and positive-definite matrix. This has two advantages. First, the search direction
is always a descent direction for $J$, which is not always the case for non-convex problems. Second, symmetry and positivity enable the use of specialized linear solvers, such as the Cholesky decomposition or the conjugate gradient algorithm, that are superior to their more general counterparts in many respects.

Other software packages that treat diagnostic models as nonlinear systems of equations tend to rely on ad-hoc procedures for initializing the numerical solution process. For example, without a damping procedure in Newton's method, the iteration

can prove unstable if initialized far away from the true solution. Some packages combat this problem by using a few iterations of the more robust but slower Picard method first (Gagliardini et al., 2013). While this approach can be effective, it requires tuning the number of Picard iterations. There is no guarantee that an adequate amount for one problem will work well on another. This issue rarely appears on realistic input data, but when solving inverse problems, the intermediate guesses for the inferred field can be wildly unrealistic before converging. A forward model solver that is not sufficiently robust can crash in these extreme scenarios. By contrast, a damped Newton procedure using a line search that satisfies the Armijo-Wolfe criteria is guaranteed to converge on non-degenerate, if unrealistic, input data. The line search method that we used here is one way to achieve global convergence of Newton's method, but there are other approaches to achieve global convergence, for example homotopy continuation (Tezaur et al., 2015) or the trust region method (Bellavia and Berrone, 2007).

## 4.3 Convergence metrics

Several works in the literature have weighed the relative merits of different iterative methods for solving the nonlinear diagnostic equation (Perego et al., 2012). Few have considered the problem of when to stop iterating. The most common stopping criteria are when (1) the 2-norm of the residual is sufficiently small or (2) the relative change in the iterates is sufficiently small. Each of these approaches has problems. The residual norm depends on the discretization and does not weight all degrees of freedom proportionally, e.g. vertex and edge degrees of freedom in higher-order finite element methods. The relative change criterion, on the other hand, can suggest convergence when in fact the method has stagnated.

We can devise a convergence criterion that works equally well, independent of the discretization and the quality of the initial guess, based on the *Newton decrement* (Nocedal and Wright, 2006). Since the second derivative operator $d^2 J(u_k)$ is positive-definite, the Newton search direction $v_k$ computed from equation (35) is a descent direction for $J$:

$$dJ(u_k) \cdot v_k < 0. \tag{37}$$

The absolute value of the quantity in the last equation is defined as the Newton decrement. For $u_k$ sufficiently close to the true solution $u$, the Newton decrement roughly tells us how much we can expect the action to decrease:

$$J(u_k) - J(u) \approx \frac{1}{2} |dJ(u_k) \cdot v_k|. \tag{38}$$

We can then use the Newton decrement to decide when to stop the iteration, as described below.

As shown in equation (4), the action for most models has units of power and is the sum of dissipation due to viscosity, friction, gravitational driving, and ocean back-pressure at the terminus. The viscous and frictional terms are convex, positive functions of the velocity. The gravitational and terminus stress terms are linear in the velocity and can be of either sign. If we define the *scale functional*

$$K(u) = \text{viscous dissipation} + \text{frictional dissipation} \tag{39}$$

as only the positive parts of the action, then the convergence criterion

$$|dJ(u_k) \cdot v_k| < \epsilon K(u_k) \tag{40}$$

is independent of the discretization. The intuition behind this criterion is that the iteration is halted when the expected decrease in the action functional is much smaller than the positive part of the action itself.

We have found empirically that, with this criterion and the Newton solver implementation in icepack, the iteration usually converges to machine precision in around 8 steps. The iteration count can reach as high as 20 for exceptionally bad initial guesses for the velocity or with unphysical fluidity or friction values. We also observe the expected doubling of the number of accurate digits in the value of the action once the velocity guesses are within the convergence basin of the true solution. Other convergence criteria, such as using relative change in the velocity guesses, can terminate prematurely when the initial guess is very far outside the quadratic convergence basin.

The numerical solvers in icepack have been designed so that users who are not familiar with numerical optimization need not be confronted with a possibly bewildering array of algorithmic parameters. Consequently, sensible defaults have been chosen for the Armijo and Wolfe criteria (Nocedal and Wright, 2006), and the tolerance for the line search is chosen based on that of the outer-level Newton iteration. The Newton search direction is calculated using a direct factorization solver rather than, say, the conjugate gradient algorithm, as the use of another iterative method would introduce yet another algorithmic parameter. Advanced users who are interested in performance optimization can change these algorithmic parameters by passing extra arguments to the solve procedure.

## 4.4 Inverse solvers

The `InverseProblem` class describes what problem is being solved, while the `InverseSolver` class is responsible for carrying out the numerical optimization. There are three inverse solvers in icepack: a simple gradient descent solver, a quasi-Newton solver based on the BFGS approximation to the Hessian, and a Gauss-Newton solver. All of these classes are based around the general idea of first computing a search direction and then performing a line search. They differ in how the search direction is computed.

The gradient descent solver uses the search direction

$$\phi_k = -M^{-1}dJ(\theta_k) \tag{41}$$

where $M$ is the finite element mass matrix. Gradient descent is a popular choice because the objective functional is always decreasing along this search direction. However, the search direction can be poorly scaled to the physical dimensions of the problem at hand. This method can be very expensive and brittle in the initial iterations and often takes many steps to converge.

The BFGS method uses the past $m$ iterations of the algorithm to compute a low-rank approximation to the inverse of the objective functional's Hessian matrix; see Nocedal and Wright (2006) for a more in-depth discussion. The BFGS method converges faster than gradient descent. However, it suffers from many of the same brittleness issues in the initial iterations before it has built up enough history to approximate the Hessian inverse.

Finally, the Gauss-Newton solver defines an approximation to the "first-order" part of the objective functional Hessian (Pratt et al., 1998; Habermann et al., 2012). Each iteration of Gauss-Newton is more expensive than that of BFGS or gradient

descent because it requires the solution of a more complex linear system than just the mass matrix. The Gauss-Newton method converges fastest by far in virtually every test case we have found, in some instances by up to factor of 50.

The derivative of the objective functional with respect to the unknown parameter is calculated using the adjoint method and the symbolic differentiation features of Firedrake. The user does not need to provide any routines for the derivatives, only the symbolic form of the error metric and the regularization functional. The model object is responsible for providing the symbolic form of the action functional.

### 4.5   Hybrid model

The hybrid flow model uses several features that are available in Firedrake to better exploit the special structure of the problem. Implementing this model also required some mathematical sleight-of-hand related to the terminus boundary condition that has not appeared in the literature before. Additionally, the hierarchical structure of spectral basis functions presents an opportunity for developing fast algorithms. In all other respects the implementation of the hybrid flow model using convex optimization follows the techniques described above.

#### 4.5.1   Discretization

In order to use terrain-following coordinates, the hybrid model assumes that the geometry of the domain is an extruded mesh, where a 2D footprint mesh is lifted into 3D. Firedrake includes support for creating extruded meshes by calling the function `ExtrudedMesh` on the 2D footprint (Bercea et al., 2016; McRae et al., 2016). The cells of an extruded mesh are triangular prisms instead of the more common tetrahedra used for general 3D meshes. Not every 3D domain can be described by extruding

a 2D domain, but the geometry of most glacier flow problems can.

The geometric correction factor in equation (19) for gradients in terrain-following coordinates can easily be represented in UFL. By defining a wrapper around the UFL `grad` function, the code to define the action functional in terrain-following coordinates is only slightly more complex than in Cartesian coordinates.

For problems defined on extruded geometries, Firedrake includes support for tensor product elements, which includes using

different bases in the horizontal and vertical directions (McRae et al., 2016). Tensor product elements are defined in Firedrake by passing the extra keyword arguments `vfamily`, `vdegree` to the constructor for a function space. In our case, we used the usual continuous Galerkin basis in the horizontal and Gauss-Legendre elements in the vertical. To select the Legendre polynomial basis, the user passes the keyword argument `vfamily='Gauss-Legendre'` or `'GL'` for short to the constructor for the function space.

Extruded meshes and tensor product elements are available in Firedrake but not in FEniCS. Other general-purpose finite element modeling packages that support tensor product elements include deal.II and nektar++ (Bangerth et al., 2007; Cantwell et al., 2015). Like most other packages in this domain, deal.II and nektar++ are written in C++, whereas our goal for icepack was to have both the core and the user interface in Python.

### 4.5.2 Ocean boundary condition

Our approach for implementing a hybrid flow model works completely seamlessly but for one important detail. The backpressure from ocean water at the calving front of a marine-terminating glacier is not a smooth function of depth. The pressure is 0 above the water line and linearly increasing below it:

$$\text{backpressure power} = \int\limits_{\Gamma} \int\limits_{0}^{1} \rho_W gh(\zeta_{sl} - \zeta)_+ \, d\zeta \, d\gamma, \tag{42}$$

where $\zeta_{sl}$ denotes the relative depth to the water line and the subscript $+$ denotes the positive part of a real number. Were we to use the standard asssembly procedure in Firedrake to evaluate this integral, we would get an inaccurate result due to an insufficient number of integration points. The resulting velocity solutions are then wildly inaccurate due to the mis-specification of the Neumann boundary condition. A blunt solution to this problem would be to pass an extra argument to the Firedrake form compiler that specifies a much greater integration accuracy in the vertical for this term. This fix reduces the errors in the velocities, but it does not eliminate them completely and it incurs a large computational cost.

We instead implemented a routine that symbolically calculates the Legendre polynomial expansion of the function $(\zeta_{sl} - \zeta)_+$ with respect to the parameter $\zeta_{sl}$ using the package SymPy (Meurer et al., 2017). The symbolic variables for $\zeta$ and $\zeta_{sl}$ used in the SymPy representation of the polynomial expansion are then substituted for equivalent symbolic variables in Firedrake/UFL using the SymPy object's `subs` method. The Legendre polynomial approximation to this function only converges linearly as the number of coefficients is increased, since the the function is continuous but not smooth, and the approximation exhibits noticeable ringing artifacts at high degree. While the approximation itself is not very accurate, the calculated value of the integral in equation (42) is exact because of the orthogonality property of Legendre polynomials. Stated another way, the residuals in the approximation are large, but they integrate to 0 when multiplied by any Legendre polynomial up to the number of vertical modes. An example of the pressure approximations using linear, quadratic, and cubic Legendre polynomials are shown in figure 1.

The exact symbolic integration approach is both faster and more accurate than using a large number of quadrature points. The same technique could be used to exactly calculate the ocean backpressure for any model, say the full Stokes equations, using terrain-following coordinates together with a Legendre polynomial expansion in the vertical.

### 4.6 Performance

Icepack largely inherits the performance capabilities of the Firedrake package, for which we refer to the benchmarks in Rathgeber et al. (2016). Firedrake is in turn built on PETSc, which includes a rich suite of nonlinear solvers and preconditioners that have been demonstrated to scale up to hundreds of processors (Balay et al., 2019). Firedrake also includes special features for defining sophisticated solvers and preconditioners that take advantage of problem-specific structure (Kirby and Mitchell, 2018). Icepack exposes these tools through the interface of the flow solver objects, so users can select any solver and preconditioner from PETSc.

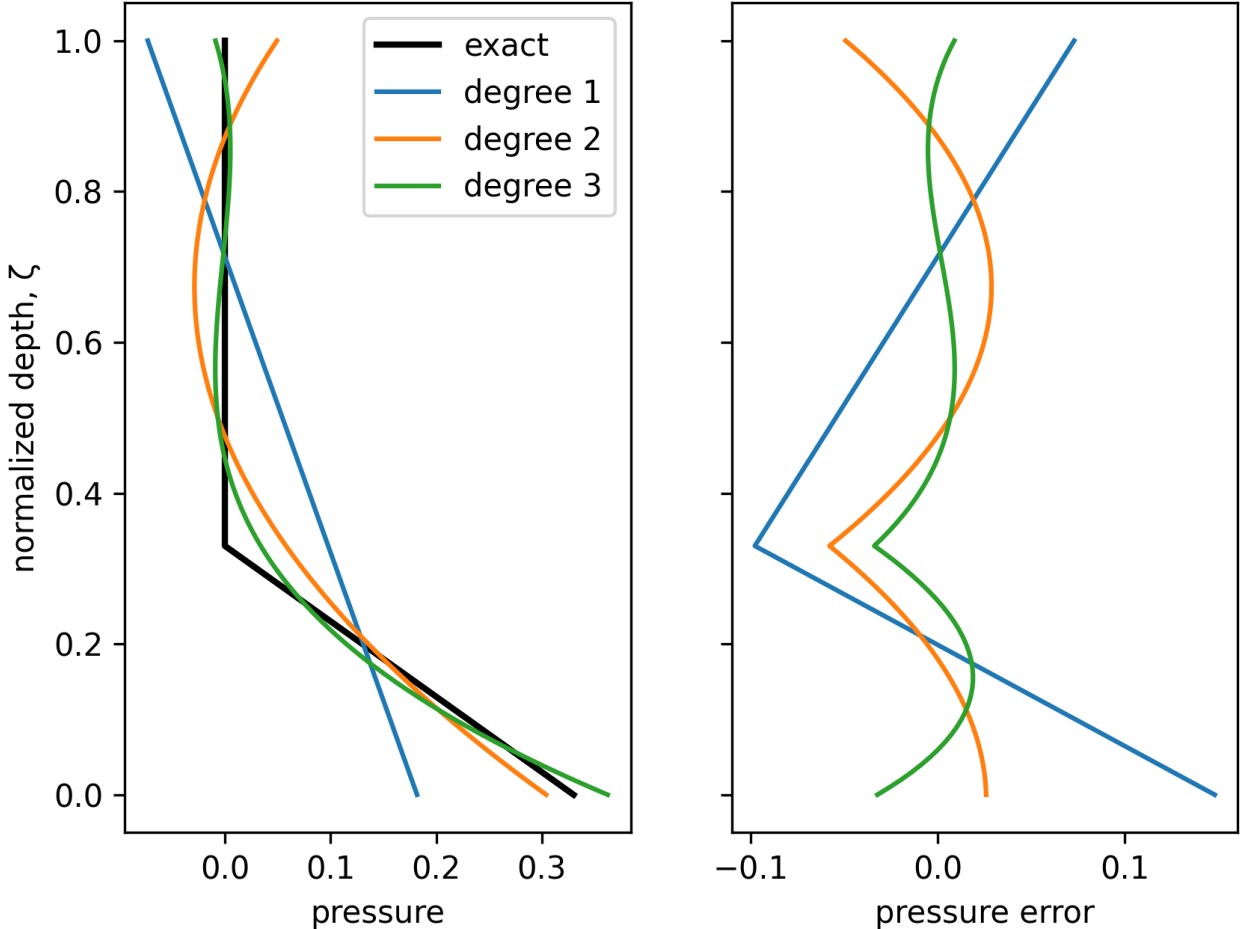

**Figure 1.** The normalized ocean pressure ($p_w/\rho_W g$) and Legendre polynomial approximations of several degrees (left), and the residuals of the approximation (right). For this particular example, the waterline is at $\zeta = 1/3$, which would be representative of a glacier grounded on a higher moraine. The moments of each of the residuals up to the approximation degree are all zero.

We have mainly developed icepack for process-scale studies of individual glaciers or drainage basins. For the demonstrations presented below, nearly all simulations run in a matter of minutes to hours on a single core. We have used sparse LU factorization to solve linear systems for many problem instances in order to eliminate the linear solver as a possible failure mode. Defaulting to a robust solution method is especially important for onboarding novice users who may not be familiar with different iterative linear solvers and preconditioners. Larger problems, such as continental-scale modeling, will require solving the diagnostic equations using the conjugate gradient method with an appropriate preconditioner to achieve parallel scalability. The particular structure of the problems we solve may be useful in choosing a preconditioner. For example, a rudimentary

preconditioner for the hybrid model system could use the degree-0 model as the coarse space in a multigrid-type approach. These optimizations will be the subject of future work.

## 5 Demonstrations

The following demonstrations aim to show the capabilities of icepack on both synthetic and real problems. The key features of icepack that these demonstrations aim to highlight are the variety of different physics models implemented and the flexibility of the components of these physics models.

### 5.1 MISMIP+

As a first test case for icepack, we ran the first experiment from the Marine Ice Sheet Model Intercomparison Project version 3 (MISMIP+). The parameters and geometry for this experiment can be found in Asay-Davis et al. (2016). The MISMIP+ experiment has three phases. First, the model must find a steady state marine ice sheet with a fixed accumulation rate and no submarine melting. Next, submarine melting with a given depth-dependent parameterization is applied for 100 years. The increased melt thins the ice ice shelf and initiates a retreat of the glacier grounding line. Finally, submarine melting is turned off for at least 100 years, optionally longer. The grounding line then readvances, but not as far as its original position.

The original intercomparison project specified that participants could use the Weertman sliding law (equation (28)) as well as two other sliding laws that transition to a more plastic rheology at high sliding speeds. The first alternative sliding law consists of Weertman sliding until the stress reaches a critical value, at which point the constitutive relation transitions to perfect plasticity. The second alternative is the Schoof sliding law (Schoof, 2005):

$$\tau_b = -\frac{C|u|^{1/m} \cdot \tau_c}{(C^m|u| + \tau_c^m)^{1/m}} \frac{u}{|u|}, \tag{43}$$

where the critical stress $\tau_c$ is a certain specified fraction of the water pressure in the subglacial hydrological system. Sliding laws in icepack are not expressed directly, but rather as the the derivative of an action functional. To implement the Schoof sliding law we need to know the antiderivative of equation (43). We found the antiderivative of this expression using a computer algebra system, but the result includes hypergeometric functions. The Unified Form Language has several transcendental functions (sine, cosine, exponential, etc.) but it does not currently support hypergeometric functions. Instead, we implemented a sliding relation that exhibits the important features of the Schoof law, i.e. it behaves like an $m = 3$ power law at low sliding speeds and $m = \infty$ at high sliding speeds, but which is more tractable algebraically. Knowing the critical stress $\tau_c$ and the friction coefficient $C$, which has units of stress $\times$ speed$^{-1/m}$, we can define a *critical speed* $u_c$ as

$$u_c = C^{-m}\tau_c^m. \tag{44}$$

The power dissipation density for the Schoof-type sliding law that we use is

$$P = \tau_c \left\{ \left( u_c^{\frac{1}{m}+1} + |u|^{\frac{1}{m}+1} \right)^{\frac{m}{m+1}} - u_c \right\} \tag{45}$$

Figure 2 shows a comparison of the original Schoof sliding law and the sliding law that arises as the derivative of the functional in equation (45). The two have the same asymptotic behavior when the speed is much smaller or much larger than the critical speed; they differ in a relatively small range around the critical speed. The relative difference in basal shear stress between the two sliding laws is less than 10% throughout the entire range. Although figure 2 shows a comparison of both sliding laws with the same value of the critical speed, by using different values, the agreement between our sliding law can be brought into much closer agreement with the Schoof law. Finally, the Schoof law itself is phenomenological. A different sliding law with the right asymptotic behavior is no more or less valid.

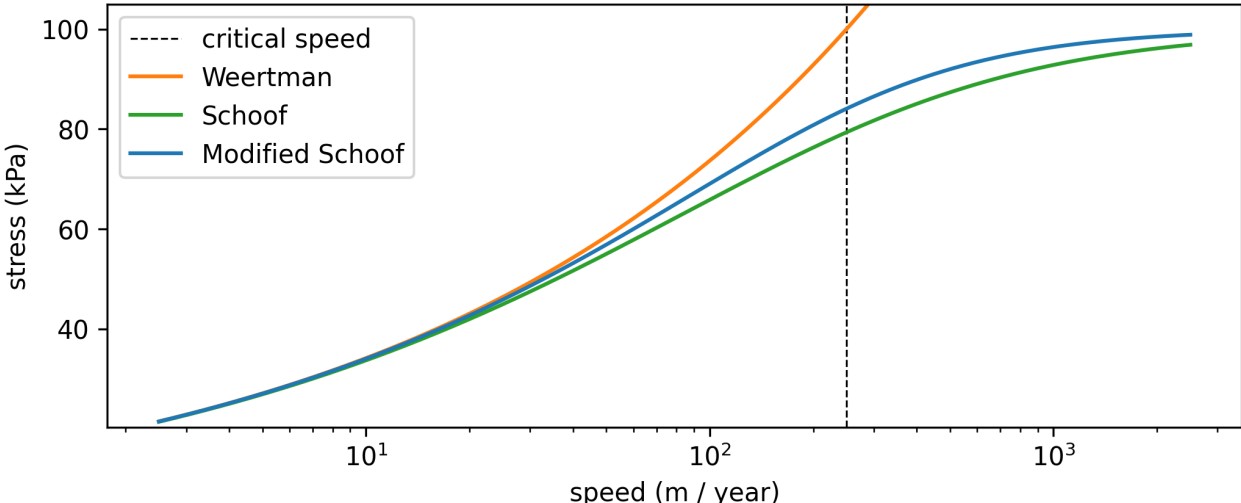

**Figure 2.** The Weertman, Schoof, and modified Schoof sliding law of equation (45). The critical speed and shear stress are $u_c = 250$ m/year and $\tau_c = 100$ kPa.

To change the sliding law, users only need to pass one function or the other to the model object at initialization. (See *Code and data availability* below for the source code.) Users do not need to implement a subclass that overrides some parent method. This approach would be idiomatic in C++, but it requires knowledge of object-oriented programming that a non-expert might lack.

Figure 3 shows the thickness and ice speed after spinning up the MISMIP+ geometry and input data to steady state with the sliding law described above. The spin-up used used degree-1 basis functions for both thickness and velocity. To get a high-resolution estimate for the steady state, the spin-up started from a relatively coarse resolution to propagate out most of the transient signal. Then the mesh was successively refined and spun up again to propagate out the remaining high-wavenumber transients. This process was repeated three times. The net result is that most of the spin-up is done at relatively little computational cost. Figure 4 shows the total ice mass during the retreat and readvance phases of the experiment. The initial response to turning on melting is very rapid but then becomes roughly linear with time. When the high melt is turned off at the 100-year mark, the ice readvances, but the rate is much slower than the rate of decrease when melt was on. This asymmetric response

is typical and expected from ice physics; the rates broadly agree with the reference results computed with BISICLES in the original experimental specification (Asay-Davis et al., 2016).

We also ran the experimental setup to steady state using the hybrid flow model with vertical basis functions up to degree 2. This is the most minimal set of vertical basis functions that can resolve plug flow and the stress boundary conditions at the ice base and surface. The ratio of basal velocity to surface velocity is shown in figure 5. The areas with the most significant

vertical deformation are at the inflow boundary and where the troughs at the side walls are steepest. Otherwise, the sliding ratio is above 0.8 throughout almost the entire domain.

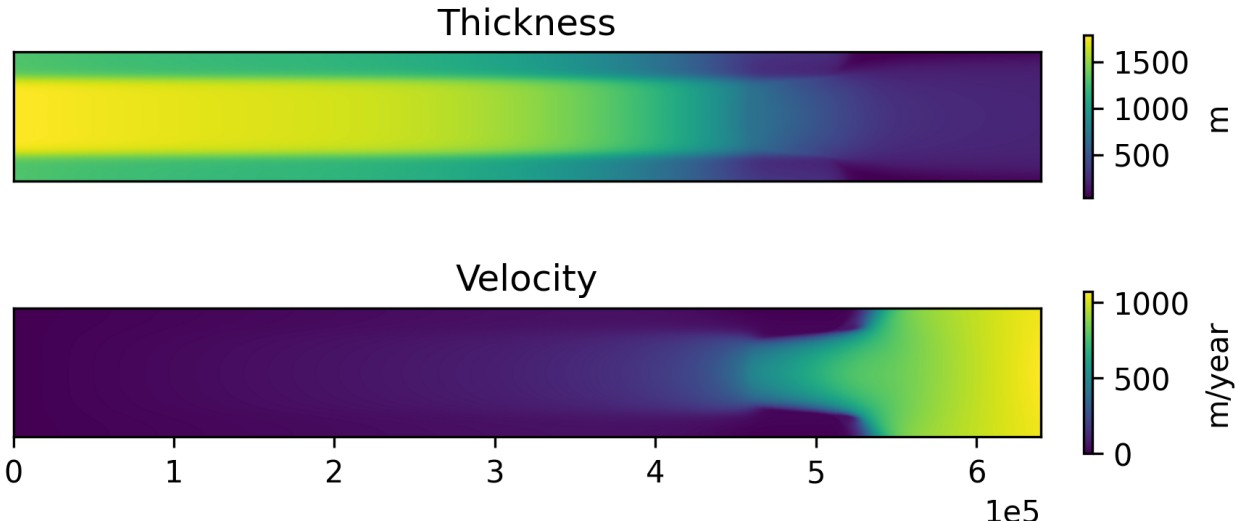

**Figure 3.** The steady state thickness and velocity of the MISMIP+ experimental setup with the modified Schoof sliding law of equation (45).

Using variational principles to express all constitutive laws is less flexible than specifying the sliding law directly and this is a distinct disadvantage. We were nonetheless able to implement a sliding law that exhibits the important characteristics of the Schoof sliding law. Asay-Davis et al. (2016) also suggest using a sliding law that transitions sharply to exact plasticity above

685 the critical speed. Expressing this sliding law in UFL requires a conditional in the velocity and is thus no longer differentiable, causing the forward solver to crash. The numerical advantages of using variational principles are so great, however, that we view this tradeoff as acceptable.

### 5.2 Synthetic ice sheet

As a demonstration of the shallow ice approximation model, we ran an experiment inspired by Kessler et al. (2008). In this

work, the authors coupled an ice flow model to simple models for bed-erosion, calving, and glacial isostatic adjustment (GIA) to investigate the formation of deep fjords that are characteristic of coastlines on Baffin Island, Greenland, and British Columbia,

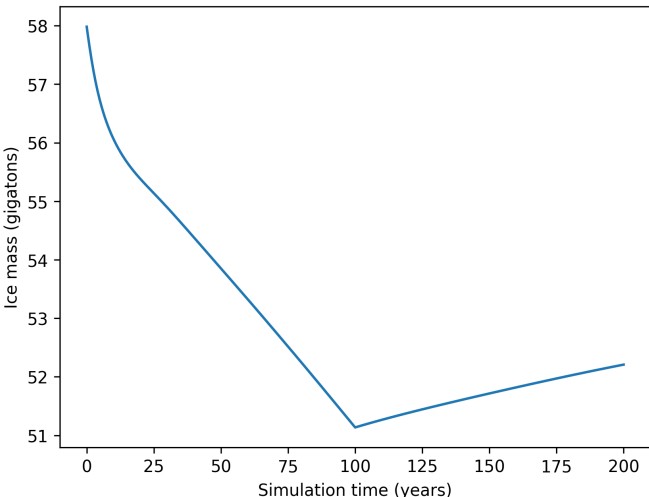

**Figure 4.** Total ice mass history during retreat (first 100 years) and readvance (second 100 years) of MISMIP+ experiment.

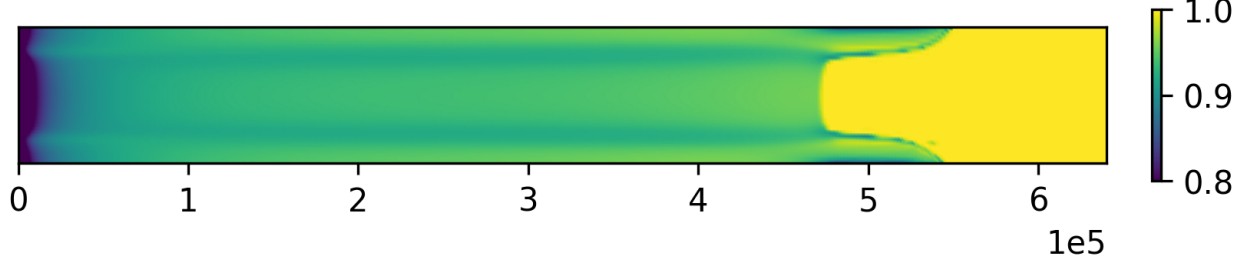

**Figure 5.** Ratio of basal velocity to surface velocity in steady state of MISMIP+ scenario computed with hybrid model.

among others. We emulated their domain and bedrock geometry – a plateau surrounded by a ridge punctuated by four valleys – without also simulating erosion or GIA.

Figure 6 shows the results of this computational experiment. We initialized the experiment with a simple but unrealistic ice thickness. We then evolved the ice sheet without climate forcing for 500 years. The ice sheet relaxes very rapidly in the first 200 years, but after this changes are much slower as ice must be funneled through one of the four narrow valleys. The resulting velocity pattern is similar to that of Kessler et al. (2008), with the highest velocities where ice flows out from the valleys. In and upstream of the valleys, the surface is drawn down due to the elevated export of ice.

The diagnostic model used in this demonstration is computationally cheap enough that it can be used to simulate ice sheets over several millenia in a matter of minutes on a desktop. A key feature of icepack is the ability to choose between many different diagnostic models. Users are free to decide what model works best for the spatial and temporal scales of their problem,

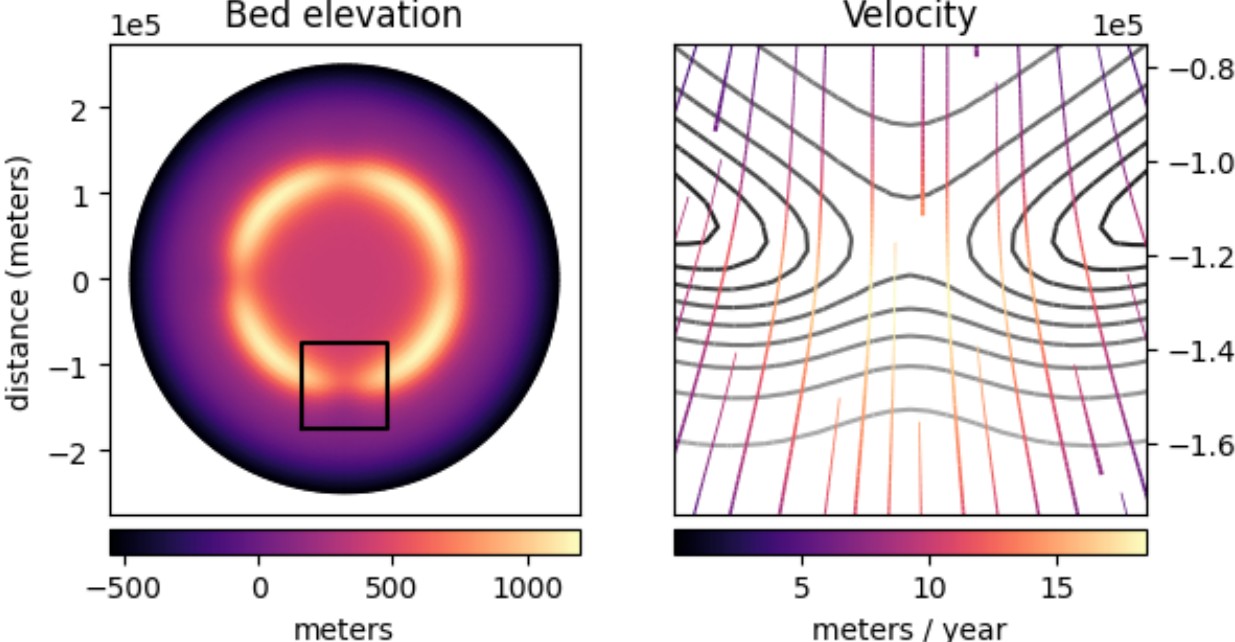

**Figure 6.** Synthetic ice sheet simulation. (a) The bed elevation profile consists of a ring of mountains with valleys of varying depth surrounding a flat plateau where the ice sheet is initialized. The black square outlines the domain of subplot (b) which shows the velocity of the ice as it passes through the southernmost mountain pass, with contours of the bed elevation shown in greyscale.

how accurately they need to solve this problem to produce useful results, and what computational resources they are willing to devote.

### 5.3 Synthetic ice shelf

We simulated the evolution of a synthetic ice shelf towards steady state and coupled it with the damage transport model described in §2.6. The geometry of the ice shelf consists of the intersection of two circles, one with a larger radius and offset center, designed to roughly mimic the shape and size of real ice shelves. The radius of the whole shelf is 200km. Four ice streams flow into the shelf with varying speeds and a prescribed inflow thickness. In the first phase of the experiment, the ice shelf is propagated to steady state without damage for 200 years. After this time interval, the flux imbalance is less than

1% of the influx. In the second phase, damage transport is turned on and coupled to the ice velocity. An interesting feature to observe in the approximate equilibrium damage field is that the highest values occur between the streams and not within them. Additionally, the spacing between the streams changes as a result of adding damage. The final ice thickness, velocity, strain rate, and damage are shown in figure 7.

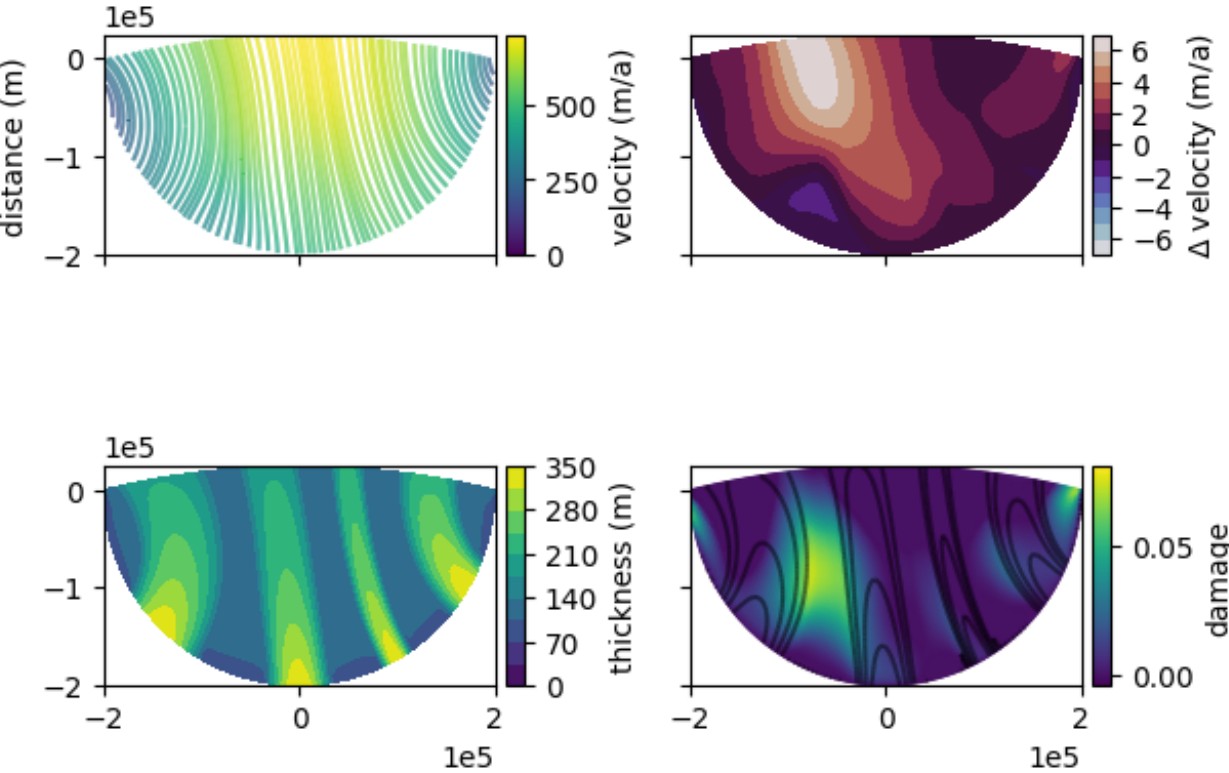

**Figure 7.** Results of ice shelf damage simulation: (a) ice velocity without damage, (b) change in ice speed with and without damage, (c) thickness, and (d) damage field.

## 5.4 Larsen C Ice Shelf

To demonstrate the inverse solver, we will estimate the rheology of the Larsen C Ice Shelf in the Antarctic Peninsula from observational data. Several recent studies have focused on Larsen C because it may be unstable in the warmer climate of the coming decades. From January to March of 2002, the neighboring Larsen B Ice Shelf disintegrated due to surface melt pond-induced fracture (Banwell et al., 2013). This mechanism might also lead to the breakup of Larsen C in the future. One of the key factors affecting the stability of ice shelves is the presence of marine ice – seawater that freezes onto the base of an ice

shelf – in the suture zones where two flow units meet (Kulessa et al., 2014). Marine ice is warmer than meteoric ice and usually includes brine pockets, which are discernible in radio echo sounding measurements as the absence of reflection from the ice shelf base (Holland et al., 2009). This warmer and impurity-laden ice is more ductile and thus should be less prone to fracture than cold and brittle meteoric ice. Ocean models predict that marine ice forms under Larsen C as well (Holland et al., 2009). By estimating the material rigidity of an ice shelf, we can constrain where marine ice may be forming.

### 5.4.1 Data

We used the InSAR phase-based ice velocity map from Mouginot et al. (2019). This dataset has nominal errors over the Larsen Ice Shelf on the order $1 - 7$ meters per year. We used the recently-released BedMachine map of the thickness of Antarctica, which takes advantage of newly available remote sensing data (Morlighem et al., 2019).

Existing work on glaciological inverse problems uses the mismatch between the computed and observed velocity fields as part of the objective functional. The effect of thickness errors is studied largely through a posteriori validation (Joughin et al., 2004; Larour et al., 2005). Errors in thickness or surface slope can be large enough that it might be impossible to fit the velocity measurements to the degree that statistical theory predicts (MacAyeal et al., 1995). The velocity measurements themselves might have significant outliers, in which case using the usual weighted sum of squared misfits as an error metric will give poor results. For these reasons, some studies have explored alternative objective functionals (Morlighem et al., 2010). We have opted to use the regularized $L^1$-type error metric

$$J(u) = \int_\Omega \left( \sqrt{\frac{|u - u^o|^2}{\sigma^2} + \gamma^2} - \gamma \right) dx. \tag{46}$$

This error metric approaches the usual weighted sum of squared errors as $\gamma \to \infty$, and approaches the sparsity-promoting $L^1$ error metric as $\gamma \to 0$. For finite, positive values of $\gamma$ this error metric is robust to non-normality or a small fraction of outliers (Barron, 2019).

### 5.4.2 Parameterization

The rheology parameter of an ice shelf is strictly positive. The optimization algorithm, however, can explore unphysical regions of parameter space without some a priori constraints. Rather than try to solve an inequality-constrained problem, most studies in glaciology instead re-parameterize the problem in terms of some auxiliary field in such a way that the rheology is manifestly positive. In this case we use the parameterization

$$A = A_0 e^\theta \tag{47}$$

and estimate $\theta$. The inverse solver calculates the derivative of the objective functional symbolically and is thus agnostic to the particular parameterization. The only information that the user needs to pass is the name of the arguments to the forward model representing the parameter and the observed field so that these can be passed by keyword.

### 5.4.3 Results

The inferred parameter field $\theta$ is shown in figure 8. The algorithm detects areas of much lower ice rigidity around highly damaged ice. This feature is especially apparent around the large rift emanating from the Gipps Ice Rise, as well as the crevassed areas upstream. We find other areas of low rigidity in the suture zones where two flow units converge and where marine ice tends to form, releasing heat to the ice shelf, exactly as observed in Holland et al. (2009). Finally, the inferred

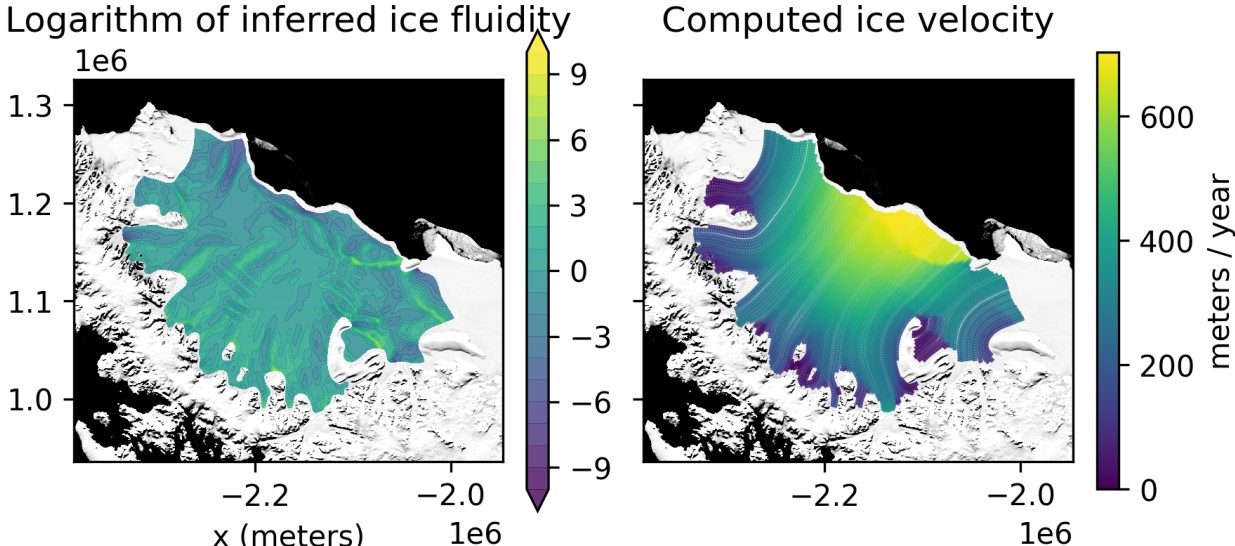

**Figure 8.** Left: Inferred fluidity parameter $\theta$. Lower values indicate more deformable ice. Right: stream plot of ice velocity computed from this fluidity parameter. Background image is the MODIS Mosaic of Antarctica (Scambos et al., 2007; Haran et al., 2014), courtesy of NASA NSIDC DAAC.

rheology field reproduces features that have already been found in previous studies of Larsen C Ice Shelf (Khazendar et al., 2011).

The final value of the model-data misfit from equation (46) matches the value we would expect if the velocity errors actually came from this probability distribution. In a separate computational experiment, we used the older bedmap2 ice thickness (Fretwell et al., 2013). We were unable to achieve the same model-data misfit using bedmap2 at the same grid resolution with any regularization parameter.

## 6  Usability

One of the main goals for icepack is to create a tool that is accessible to researchers who might not be experts in scientific computing. Previous work on numerical modeling of glacier flow has focused largely on the technical details of the models themselves – does a given solver converge with the accuracy expected from finite element theory; does it scale to large numbers of processors; can models accurately predict grounding line retreat, etc. For graduate students or other researchers who are not experts in computational physics, the difficulty of learning to use a particular software package may be more of a rate-limiting factor than the speed or efficiency of that software.

The field of *human-computer interaction* (HCI) asks how we can design software that is easier to learn and use effectively. In the following, we will describe some of the design choices in icepack and how they relate to what HCI researchers call the

*cognitive dimensions of notations*. Green and Petre (1996) introduced this concept to assess the usability of visual programming languages, but the criteria they laid out in their study have been used to analyze software systems across many disciplines.

**Consistency: After a user learns part of the software, can they guess the remaining parts?** Each of the model objects in icepack is a class with a method ending in `solve` that takes in keyword arguments for the various input fields and options for things like boundary conditions. Users already familiar with, say, the `IceStream` class can then use the `HeatTransport` class under the assumption that the input fields – the current temperature $T$, ice velocity $u$, and basal melt rate $m$ – are passed as keyword arguments with the same name as the fields themselves. Consistency obviates the need for repeatedly consulting the documentation or examples once users are already familiar with the software.

**Progressive evaluation: How easily can users get feedback during their use of the software?** Progressive evaluation is the main advantage of having a user interface in an interpreted programming language such as Python as opposed to a compiled language where programs can only run in batch mode. In the early stages of the development process, any non-trivial simulation of a physical system exists as a prototype which may be non-functional or even broken. The ability to manually step through a simulation and examine the entire state in an interpreter is critical to finding errors faster. Icepack was designed to give fine-grained control over how simulations work in order to support this mode of debugging. The API provides routines for solving the diagnostic and prognostic equations. It is the user's responsibility for making the repeated calls to these routines, either in a loop or by manually iterating through one step at a time. With this responsibility comes the freedom to add in arbitrary code. This capability might be used to add sanity checks, such as printing minimum and maximum thickness or velocity values. It can also be used to get feedback on how long the simulation will take, or to save results for later visualization. Other packages support a more coarse-grained view where the user only specifies the start and end time of a simulation and has more limited options for inspecting the state of the running program.

Icepack makes extensive use of Jupyter notebooks as a form of executable documentation. Jupyter notebooks are a document format that includes code and explanatory text with typeset mathematics that runs interactively in a web browser (Kluyver et al., 2016). The contents of a notebook can executed incrementally much like running code in the Python interpreter. Most importantly, Jupyter notebooks can render and display visualizations on the fly. This enables a workflow where plotting all intermediate results serves for sanity-checking an experimental simulation.

Simulations that have been debugged can then easily be transformed into a single Python script, for example using the tool `nbconvert`, for production runs and parallel execution. In other words, while there is an interactive interface, there is also a faster batch mode interface as well.

**Abstraction gradient: What are the levels of abstraction exposed by the library? Can irrelevant details be hidden?** The API for icepack has been designed so that the users only need to decide what problem to solve and not how to solve it. Where a choice does concern more the "how" than the "what", we use a sensible default that biases for correctness rather than speed. For example, the Newton solver uses a direct factorization method to solve the linear system for the search direction because factorization requires no tuning whereas iterative methods do. A user interested in achieving greater runtime performance can pass additional keyword arguments instead specifying, say, the preconditioned conjugate gradient method. This choice

is of interest mostly to advanced users so we keep the linear solver algorithm as a default argument. In so doing, we avoid confronting novice users with options that they might not understand.

Advanced users who do wish to tune solver performance for large simulations will need some way to make choices about algorithms. For example, one might choose the GMRES iterative solver together with an incomplete LU preconditioner to solve linear systems. The solver classes, as opposed to the model classes, provide the interface for making these choices. While alternative solvers might offer faster runtime performance than direct factorization, they also requires making additional choices – how often to restart GMRES? How much fill-in to allow in the incomplete factorization? Many glaciologists do not have

the background in numerical linear algebra to know that adjusting these parameters could make the difference between solver convergence or breakdown. As another example, users might want to select between different discretization strategies for the Stokes equations. The discretization of the Stokes equations has to be chosen carefully in order to satisfy the Ladyzhenskaya-Babuška-Brezzi (LBB) conditions (Boffi et al., 2013). When using Galerkin least-squares stabilization of the weak form, the user has to pick a value of the stabilization parameter. Determining exactly what value of this parameter is necessary to

guarantee stability is a subtle problem, even more so for the kinds of highly anisotropic meshes that are commonly encountered in 3D glacier flow modeling. If the solver fails to converge, it might not be obvious even to an expert whether the problem lies with the stabilization or the aforementioned parameters of the linear solver.

## 7   Conclusions

We have presented a new software package called *icepack* for modeling the flow of glaciers and ice sheets. This package

advances the state of the art in this field by providing a platform for easily experimenting with the model physics. In this paper, we have presented three demonstrations of this feature:

1. coupling a model of ice shelf flow to a phenomenological model of damage,

2. changing the sliding law in a simulation of a marine-terminating glacier, and

3. inferring the fluidity of a floating ice shelf in a way that guarantees positivity.

The physics of how glaciers interact with their environment are not completely understood. Consequently, the ability to change components of the model is an essential feature for any tool aimed at researchers in this field.

    We have also paid special attention to how we can design this software package to be most usable for its intended audience. Relatively few works in the computational science literature draw directly from relevant work in HCI when discussing usability; see for example Hannay et al. (2009); Harris et al. (2020). We believe that this is because of two difficulties. First, the

830 degree to which usability is a rate-limiting factor for scientists is hard to quantify and likely differs widely across disciplines. Second, concretely assessing what features make software tools more or less usable is highly subjective. By contrast, measuring computational performance is much more feasible although still fraught with difficulties of its own. (This is not to say that performance optimization is easy by any means.) In working with several graduate students and postdoctoral researchers in glaciology, we have observed that usability is a substantial bottleneck for researchers at these career stages. For this reason,

we have chosen to focus explicitly on usability by drawing on the research literature in HCI. Exactly how to apply principles from HCI to maximize usability is, nonetheless, not an exact science. Our implementation may have failed to meet this goal and changes in future versions will be guided by what users find most difficult. Finally, we argue that the same design features that enhance usability for relative novices to the subject area will also enhance the productivity of expert users.

This paper has presented several physics models currently implemented in icepack along with demonstrations. Future developments will include:

1. an implementation of full Stokes flow,

2. improved physics formulations and solvers that work in ice-free areas,

3. adaptivity in time and space,

New features will be guided by feedback from icepack users and from the glaciological community at large. By providing implementations of several glacier flow models from less to more complex and by enabling users to experiment with the physics, this tool both lowers the barrier to entry for novices to numerical modeling and provides a pathway for these users to progress towards ever more sophisticated and advanced simulations.

*Code and data availability.* All code used in this repository is free and open source and all data sets used in the demonstrations are publicly available. The icepack source repository and regression testing results are at https://github.com/icepack/icepack. Icepack is released under the GPLv3 license. The git commit hash of the version of the code used for this publication is b78b0ee5, see also the Zenodo release at https://doi.org/10.5281/zenodo.1205640. The icepack documentation, user manual, and contribution guidelines are hosted at https://icepack.github.io.

The source code for the demonstrations used in this paper is hosted at https://github.com/icepack/icepack-paper, commit hash 4db3dc28. The demonstrations used glacier outlines that were hand-digitized from satellite imagery in order to generate the model domains. These outlines are kept in version control and hosted at https://github.com/icepack/glacier-meshes and the git commit hash for the version used in this publication is c98a8b75. Additional observational data are hosted at the US National Snow and Ice Data Center (https://www.nsidc.org).

*Author contributions.* DS designed and implemented icepack with contributions from JB and AH. IJ tested the model and assisted with design and debugging. All authors contributed to the demonstration codes and to writing this paper.

*Competing interests.* All authors declare that they have no competing interests.

*Acknowledgements.* We would like to thank the Firedrake development team for their support and for many helpful discussions.

*Financial support*. DRS was supported by US National Science Foundation grant #1835321 and National Aeronautics and Space Administration grant #80NSSC20K0954. JAB was supported by NSF grant #1256082. AOH was supported by NASA grant #80NSSC20K1627. IRJ was supported by NSF grants #1835321 and #1643285.

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
