# Peer review of "icepack: a new glacier flow modeling package in Python, version 1.0"

_Geoscientific Model Development, 2020_

## Referee Comment (RC1)

**Doug Brinkerhoff**

**February 2021**

**Summary**

In 'icepack: a new glacier flow modeling package in Python, version 1.0', Shapero and co-authors present a promising new ice sheet modeling framework. The framework contains mechanisms for solving both the prognostic mass balance equations for updating ice sheet geometry, as well as diagnostic solvers for approximations to the non-linear Stokes' equations. Throughout both the software and the manuscript describing it, the authors focus on ensuring usability (read-ability), a trait that is bound to make this software (and paper) frequently used. Despite its accessibility, the capabilities of icepack are already impressive, made all the more so by its explicit design prioritization of easy extensibility.

I find the manuscript to be exceptionally well-crafted, and I think that it could be published as is. That said, I offer a few suggestions, comments, and points of clarification below.

**Minor points**

- L15-18 It would be nice to have a cited example or three for each of these suggested use cases. This would help the reader identify the kinds of practical problems where icepack might fill a need.
- L74 A low aspect ratio isn't really an approximation; it's an existential fact. The approximation that the first order approximation makes is that vertical resistive stresses (or bridging stresses as they are referred to later in this manuscript) are small, pressure is hydrostatic, and bed slopes are small.
- L92 It would be useful to offer a reference regarding an anisotropic fluidity.
- L100 Not clear where the Legendre transform enters: the viscous and frictional dissipation can be read off from Eq. 4.
- Eqs. 6 and 12 While I understand that it is convenient to manipulate the action to reflect the algebraic manipulations to yield the analytical SIA solution, the break in symmetry between Eq. 6 and Eq. 12 is frustrating,

given that they both are name 'gravity', and that they should in some sense be the same regardless of which strain rates are assumed to be zero.

- L144 That the terminal potential term, Eq. 13 disappears is not obvious. This should probably be clarified, since many readers will be surprised by this.
- L152 Cite the method of manufactured solutions.
- L159 The benefit to avoiding complicated 3D meshing should not be understated, in addition to the reduction in the cost of computational solution.
- L188 This is a bit of a red herring, given that impenetrability is a natural boundary condition on the incompressibility equation. It can go right in the action principle, no extra Lagrange multiplier (besides pressure) needed.
- L367 I'm surprised by this. For challenging geometries, I've always needed to stabilize even when using implicit Euler. Are you sure that implicit Euler is unconditionally stable even for non-linear advection like this?
- L453 Many advances have occurred in the last 5 years regarding gradient descent due to its necessity for optimizing neural networks. These may yet be useful in this context if you have to deal with a large scale optimization problem where forming the Hessian becomes prohibitive.
- L459 Gauss-Newton needs a reference if you're not going to describe it here.
- Eq. 34 It's worth noting that the Schoof law is phenomenological, and was selected because it has the right shape and obeys Iken's bound. As such, if your sliding law obeys Iken's bound and looks right, then it's not any less valid.
- L580 I think that having to specify a variational principle for the sliding law is useful because it guarantees a law that is positive (semi-)definite, as it must be to be physical.
- L701–713 I enjoyed reading this paragraph, but I wonder if stabilizing the Stokes' equations is the best illustration of the point, given that icepack does not in fact have Stokes' equations implemented (although I imagine it could be done in short order). Maybe stabilizing a transport equation would be a more appropriate case?

---

## Referee Comment (RC2)

**Review of "*icepack*: a new glacier flow modeling package in Python, version 1.0" by D. Shapero, J. Badgeley, A. Hoffmann and I. Joughin**

February 18, 2021

In this manuscript, the authors introduce a new land-ice modeling software package known as *icepack*. *icepack* is written in Python on top of the Firedrake library, which uses the domain-specific Unified Form Language (UFL) and provides a high-level symbolic description of the problem to facilitate the add of new physics and/or equations. The intended user base of *icepack* is the glaciological community, in particular, glaciologists who may not have an extensive background/training in computational science. The idea is to provide a code that would be easy to use/develop by this class of prospective users. Following a description of the code, some numerical examples are presented to demonstrate the method's capabilities and accuracy.

The manuscript in question is well-written and interesting to read. Addressing usability is noteworthy and something that not enough authors in glaciology/climate science address. I personally have some qualms with some of the philosophy described by the authors, namely I worry about folks who are not familiar with numerical methods developing an application code in which a lot of what is under the code is hidden from them in some sense. In an ideal world, one would have glaciologists working with computational scientists to help them pick the right solvers, discretizations, etc., for their problem. The authors are correct that some solver options are for optimizing performance, which is secondary to getting the code/model running; but there are also solver/algorithm choices that depend very much on the physics (e.g., CG is only valid for symmetric problems) - is *icepack* designed so as to prevent the naive user from inadvertently using the incorrect default setting for their problem? I will assume it is to the extent it can be, and that the authors' argument is that the default hidden settings are likely to do less damage than some arbitrary settings a user might put in his/her input file without knowing what they are doing. Also, I realize that the reality is that many glaciologists do not have strong ties to computational scientists, and still wish to make progress in numerical modeling of land-ice; therefore, I will not focus too much on much "philosophical" perspective described above.

Another qualm I have about the paper has to do with performance - I am skeptical whether *icepack* can really be performant if one tries to run it on continental scale problems, and it is not clear to me if the code is even parallel. I think the intention may be to use *icepack* as more of a sandbox for prototyping small problems (similar to FeniCS), in which case, this is not a huge deal.

Despite the above concerns, I like this paper and see it published in GMD. I like in particular the idea of describing all the equations using the variational principle/action functional and having everything else propagate from there - not enough people do this. I do, however, feel that there is a lot of missing information in various parts of the paper, which should be filled in in the revision prior to the paper being suitable for publication. Please see below my enumerated list of questions/comments to address in the revision.

1. The authors suggest that C++ makes it inherently difficult to add new physics/PDEs (e.g., on p. 10

and p. 20), which I somewhat disagree with. One advancement that can make a C++-based code easy to add to is Automatic Differentiation (AD) - with AD, one can effectively code the weak form of the residual within a C++ code and AD will handle the rest, making it very easy to add new physics. An example is the Albany/Land Ice model (previously known as Albany/FELIX) of Tezaur et al. (`https://doi.org/10.5194/gmd-8-1197-2015`). I think it would be worth mentioning that there have been efforts like Albany/Land Ice out there to make C++-based codes more accessible to users of varied backgrounds. I agree that even an "easy-to-use" C++ code will be more difficult and more intimidating than a Python code, so I am not trying to minimize the authors' efforts at all.

2. Can the authors comment on the overhead of the symbolic descriptions/manipulations done by their framework? This sounds potentially like it would be very expensive. How does the cost compare to automatic differentiation, for example? A broader question is: is computational performance/cost a concern for users of *icepack*, or is it intended to be a "sandbox" in which performance is secondary to being able to code up something "quick-and-dirty" for initial prototyping?

3. I was a little bit confused about the reference of the FO Stokes-based model in this paper as a "hybrid model". I see that it is hybrid in the sense that you have a different discretization in the horizontal and vertical direction, but there are also hybrid ice models that use different PDEs in different domains, e.g., the ISCAL model of Ahlkrona et al. (`https://doi.org/10.1016/j.quascirev.2016.01.032`). Is the term "hybrid model" a common name for the approach in Section 2.2.3? Perhaps it is and I am not aware of it.

4. Does the hybrid model described in the paper have the same applicability as say the First Order Stokes model? Can it be used for both Greenland and Antarctica at continental scales?

5. Section 2.2.1: in my opinion, the authors do not provide sufficient justification for the penalty term, equation (7). They describe this as something that is added to smooth over artifacts - this would be needed based on the discretization, which there is little discussion of. The authors should state what order finite elements they are using - I presume they are linear, and that this is why the stabilization is needed? Why is stabilization needed only for the SIA? I think these things should be made clear.

6. Section 2.2.2: there is some imprecision here in equation (13) - you have not defined anywhere that $\Gamma$ is the boundary, and which boundary you are referring to. One can figure it out, but it is not precise. $\Omega$ is not defined either though one will assume invariably that this is an open bounded domain in 2D or 3D depending on which approximation one is looking at.

7. The boundary conditions are not discussed very rigorously systematically - the authors seem to sprinkle in some boundary conditions here and there. I think the boundary conditions need to be given for each of the models at the time the models are presented - boundary conditions are needed to complete the definition of each models.

8. Certain terms in the equations I do not believe are defined anywhere, for instance, in equation (10), there is no expression given for the strains $\dot{\epsilon}(u)$. This is one of the things I cam across that need to be made more precise.

9. Section 2.2.3: the authors comment that higher degree polynomials can be used in the vertical layer in the hybrid approach. What order is typically used?

10. Section 2.2.3: this might be a naive question, but does Glen's flow law come into the hybrid model? I was expecting to see it there, but maybe I'm missing something.

11. P. 10: the discussion here about substituting model components suggests it may be possible to use different models in different regions and couple them (a la the ISCAL method). Is this possible, or something that the authors are thinking to add to their model/code?

12. Section 2.4: This section is very incomplete. You need to give the enthalpy equation and given the Glen's law expression as well, since it is mentioned.

13. In my opinion, there is not enough discussion of the thickness equation (Section 2.1) and how it is discretized. In typical ice sheet models, this equation is used to change the ice extent - one meshes up a region of "potential" ice, and then uses the thickness to dynamically determine a mask for ice-covered regions. Do you do something like this in your model? It should be discussed for completeness. I think you maybe start to do this in Section 4.1, but it is very confusing and hard to make the connection.

14. What sort of meshes do you use in your model (in the horizontal dimension, for the hybrid one)? Structured/unstructured? Hex/tet (quad/tri)?

15. Section 3: It is not clear from the description what the inverse problem you are describing is for. Is it to obtain parameters in the model like the basal friction using observational data of e.g. surface velocity? There really needs to be more discussion here, and I personally would like to see a mathematical statement of a representative inverse problem you are solving. It would be worth citing the work Perego et al. on optimization-based inversion, if what you are doing is similar: `https://doi.org/10.1002/2014JF003181`. BTW, the basal friction has not been defined, yet it is discussed - it needs to be defined earlier, when talking about boundary conditions (which needs to be added).

16. Section 4.1: I find this section confusing. I assume you are talking about discretizing the thickness equation here - that should be made clear. I disagree with several statements in this section as well. "The simplest explicit timestepping schemes are unstable with CG finite elements" - if you are talking about CFL stability, this is not true. You need to satisfy a CFL condition which could give rise to very small time-steps but you can get the scheme to be stable. I'm also confused about the notion of SUPG as a time-stepping scheme - I think of SUPG as a finite element approach to deal with advection-dominated flow problems, for example, that does not have anything to do with time-stepping. Maybe you are referring to upwinding? In any case, I think SUPG has nothing to do with forward Euler, so the discussion about forward Euler requiring parameters is erroneous. Additionally, I don't understand the comment about implicit Euler smoothing out sharp discontinuities... I believe explicit and implicit Euler have effectively the same diffusion and dispersion properties, so there should not really be a difference between the schemes. Did the authors verify their time-stepper on a manufactured problem to ensure that it was implemented correctly?

17. Section 4.2: there is an approach discussed in Tezaur et al. (`https://doi.org/10.5194/gmd-8-1197-2015`) for dealing with bad initial conditions in a Newton solver that relies on homotopy continuation that would be worth citing. It is an alternate to the approach you describe that lets you get away with not doing a line search for Newton. By the way, it should be no surprise that Newton is not converging

without a line search - in general Newton is not guaranteed to converge from an arbitrary initial guess without the line search.

18. Section 4.3, lines 411-412: there are actually ways to construct weighted norms to deal with the issue of DOFs having different orders of magnitude for the purpose of convergence.

19. Section 4.4: Again, it is not clear to me what is your inverse problem. You need to state this explicitly so it is clear.

20. Section 4.5.1: you talk about problems defined on "extruded geometries" - do you ever use non-extruded geometries? It has been shown in various references that there can be numerical problems for land-ice solvers that do not use extruded geometries, e.g. Tezaur et al.

21. Section 4.5.2: I think this section needs to be made earlier, and other BCs need to be added to that discussion.

22. Section 4.6: The authors mention running their code on 1 core. Is the code parallel - can it be run on multiple cores? Are there any hope for performance portability of the code to take advantage of emerging HPC architectures, e.g., GPUs?

23. Section 4.6: Can you please clarify what you mean by the following statement? "Large problems, such as continental-scale modeling, will require more sophisticated and possibly problem-specific approaches". I'm wondering in particular about the problem-specific approach part. There are models like first order Stokes that can be used at the continental scale and they are not really problem specific.

24. p. 20: I don't understand why you need to create an analytical expression of (32) using special functions. Is this something specific to your framework, which requires expressions in a certain form for the symbolic representation?

25. It's great that you have executable documentation in something easy-to-use such as Jupyter notebooks! (no need to address this comment)

26. p. 29: are you considering putting in the first-order Stokes//Blatter Pattyn model into your code framework?

27. The methods do not discuss their code development/testing stance on *icepack*. How do users contribute to the code - through pull requests? Is there regression/performance testing? Continuous integration testing? These are all really important, especially if you have non-experts contributing to the code!

Minor comments:

- "UFL" is not defined in the abstract.

- p. 4, line 90: $A$ is also called the "flow factor". I would mention here that it is usually a function of the temperature, which comes from a different equation.

- p. 5, line 125: should be "checking", not "check".

- p. 5, line 126: I think it should be "Bueler" not "Beuler", if I'm thinking of the right person.

- p. 5, line 147: change "we verified the correctness of the ice shelf model" to "we verified the correctness of our implementation of the ice shelf model".

- p. 10, line 270: change "we'll" to "we will".

---

## Author Comment (AC1)

**Response to review #1**

Thank you Doug for this helpful review! Original comments are in blue, our responses in black.

**Summary**

In 'icepack: a new glacier flow modeling package in Python, version 1.0', Shapero and co-authors present a promising new ice sheet modeling framework. The framework contains mechanisms for solving both the prognostic mass balance equations for updating ice sheet geometry, as well as diagnostic solvers for approximations to the non-linear Stokes equations. Throughout both the software and the manuscript describing it, the authors focus on ensuring usability (readability), a trait that is bound to make this software (and paper) frequently used. Despite its accessibility, the capabilities of icepack are already impressive, made all the more so by its explicit design prioritization of easy extensibility. I find the manuscript to be exceptionally well-crafted, and I think that it could be published as is. That said, I offer a few suggestions, comments, and points of clarification below.

**Minor points**

**L15-18** It would be nice to have a cited example or three for each of these suggested use cases. This would help the reader identify the kinds of practical problems where icepack might fill a need.

We've added one or two references for each.

**L74** A low aspect ratio isn't really an approximation; it's an existential fact. The approximation that the first order approximation makes is that vertical resistive stresses (or bridging stresses as they are referred to later in this manuscript) are small, pressure is hydrostatic, and bed slopes are small.

Changed to: "The first-order model is based on an asymptotic expansion of the Stokes equations in the ratio of the ice thickness to a typical horizontal length scale. The aspect ratio of most glacier flows is on the order of 1/20 or less, although there are some exceptions. For example, the main trunk of Jakobshavn Isbrae in Greenland flows through a very deep and narrow trough with an aspect ratio closer to 1/5."

L92 It would be useful to offer a reference regarding an anisotropic fluidity.

Added a reference to Gillet-Chaulet et al. 2006.

**L100** Not clear where the Legendre transform enters: the viscous and frictional dissipation can be read off from Eq. 4.

Other reviewers didn't find this digression useful so we've removed it.

**Eqs. 6 and 12** While I understand that it is convenient to manipulate the action to reflect the algebraic manipulations to yield the analytical SIA solution, the break in symmetry between Eq. 6 and Eq. 12 is frustrating, given that they both are name 'gravity', and that they should in some sense be the same regardless of which strain rates are assumed to be zero.

These modules were implemented by two different authors, the SSA / hybrid models by myself (DRS) and the SIA model by Jessica Badgeley (second author) as a project for her PhD thesis and to learn more about finite element analysis. We recognize that there's a discrepancy here and this will be corrected in future releases of the package.

**L144** That the terminal potential term, Eq. 13 disappears is not obvious. This should probably be clarified, since many readers will be surprised by this.

Expanded the explanation to: "Additionally, the terminal stress term of the action disappears after applying integration by parts to the gravity term to shift the gradient of the surface elevation over onto the velocity."

L152 Cite the method of manufactured solutions.

Added a citation to Roache 2002, Code verification by the method of manufactured solutions.

**L159** The benefit to avoiding complicated 3D meshing should not be under-stated, in addition to the reduction in the cost of computational solution.

3D meshing is something to be avoided, but we make a big deal elsewhere in the paper about using extruded meshes. With extruded meshes you only need to do 2D meshing, for which there are algorithms with much better guarantees, but solving 3D problems like the Stokes equations is still possible. We chose not to add any text here as it would then contradict or obscure some of the points we make later.

**L188** This is a bit of a red herring, given that impenetrability is a natural boundary condition on the incompressibility equation. It can go right in the action principle, no extra Lagrange multiplier (besides pressure) needed.

I don't think this is entirely correct. Imposing zero velocity at the boundary is easy, but imposing no normal flow or a fixed normal flow together with friction along the opposite directions is much more challenging. When the boundary is flat, you can set one component of the velocity to 0. But if the finite element basis for the velocity has degrees of freedom that are located at the mesh vertices, you have to confront the fact that the unit outward normal vector isn't uniquely defined at the mesh vertices. In the next sentence we cited a paper from the Elmer/Ice group where they devised an ad-hoc (but still very effective) scheme for defining the normal vector at mesh vertices and thus imposing this boundary condition. You can also do it with Nitsche's method for linear problems but to my knowledge no one has figured that out for power-law fluids. I think that the natural boundary condition you're thinking of in most weak formulations of the Stokes equations is that the average of the pressure is zero.

**L367** I'm surprised by this. For challenging geometries, I've always needed to stabilize even when using implicit Euler. Are you sure that implicit Euler is unconditionally stable even for non-linear advection like this?

See Donea and Huerta, *Finite element methods for flow problems*, chapter 3, sections 4-6. This section was also written very early on but became out of date by the time we submitted the paper. Since then, we added an implicit version of the Lax-Wendroff scheme, which has better stability properties and higher order accuracy. The text has been amended to reflect this change.

We did have similar experiences to what you describe when preparing the MISMIP+ test case for this paper. The melt phase produces very high ablation rates concentrated in a small region right near the grounded line. In our initial setup, the mesh we used didn't adequately resolve this feature. The finite element interpolation errors can then have an oscillatory pattern that, while not directly amplified by the implicit Euler timestepping scheme as such, still persists and gives a nonsense solution. This behavior manifests even at timesteps substantially below the CFL timestep. High-amplitude oscillatory garbage in the thickness field can then result in unrealistically large driving stresses and crash the nonlinear solver for the velocity. The extra terms from the Lax-Wendroff method help to diffuse out these oscillatory features. Both schemes give good results when the mesh is sufficiently refined; the Lax-Wendroff scheme just require less refinement. So this could be more an issue of spatial resolution than stability of the timestepping scheme. You're right that saving users from having to think about the CFL condition doesn't completely alleviate all the difficulties and we state as much in the next sentence.

**L453** Many advances have occurred in the last 5 years regarding gradient descent due to its necessity for optimizing neural networks. These may yet be useful in this context if you have to deal with a large scale optimization problem where forming the Hessian becomes prohibitive.

One hard criterion we have is that the method needs to naturally map from the dual of the parameter space (where the gradient lives) back to the parameter space itself. Neglecting to do so often results in unspeakable horrors like the vertex degrees of freedom converging at a different rate than the edge degrees of freedom under mesh refinement. See Schwedes et al. 2017, *Mesh Dependence in PDE-Constrained Optimisation*. Using the Hessian or an approximation does that. The acceleration tricks for first-order optimization methods are really amazing but I've yet to find a nice way to adapt them to problems posed over Sobolev spaces. By contrast, I know that I can scale second-order methods to larger problems than we're solving now by using matrix-free application of the Hessian and coming up with better preconditioners.

L459 Gauss-Newton needs a reference if you're not going to describe it here.

We added a reference to Pratt et al. 1998 which, although focused on applications in seismology, I think does a better job describing it than any other reference.

**Eq. 34** It's worth noting that the Schoof law is phenomenological, and was selected because it has the right shape and obeys Iken's bound. As such, if your sliding law obeys Iken's bound and looks right, then it's not any less valid.

This was stated around line 586 but it makes more sense to say that earlier in the text. We've moved the statement accordingly.

**L580** I think that having to specify a variational principle for the sliding law is useful because it guarantees a law that is positive (semi-)definite, as it must be to be physical.

I'm not completely sure what you mean. If the basal shear stress has the wrong sign this is arguably just as easy to check by looking at the action functional as it is a nonlinear system of equations. If you mean that the action has to be convex then this isn't true, the sliding law could be rate-weakening, but then there might be multiple steady states.

**L701-713** I enjoyed reading this paragraph, but I wonder if stabilizing the Stokes equations is the best illustration of the point, given that icepack does not in fact have Stokes equations implemented (although I imagine it could be done in short order). Maybe stabilizing a transport equation would be a more appropriate case?

This would indeed be a stronger point if we actually had a Stokes solver (it's in the works now) but the example was just to be illustrative. No change to the text.

---

## Author Comment (AC2)

**Response to review #2**

**Summary**

Thank you for this helpful review! Original comments are in blue, our responses in black.

In this manuscript, the authors introduce a new land-ice modeling software package known as icepack. icepack is written in Python on top of the Firedrake library, which uses the domain-specific Unified Form Language (UFL) and provides a high-level symbolic description of the problem to facilitate the add of new physics and/or equations. The intended user base of icepack is the glaciological community, in particular, glaciologists who may not have an extensive background/training in computational science. The idea is to provide a code that would be easy to use/develop by this class of prospective users. Following a description of the code, some numerical examples are presented to demonstrate the method's capabilities and accuracy. The manuscript in question is well-written and interesting to read. Addressing usability is noteworthy and something that not enough authors in glaciology/climate science address. I personally have some qualms with some of the philosophy described by the authors, namely I worry about folks who are not familiar with numerical methods developing an application code in which a lot of what is under the code is hidden from them in some sense. In an ideal world, one would have glaciologists working with computational scientists to help them pick the right solvers, discretizations, etc., for their problem. The authors are correct that some solver options are for optimizing performance, which is secondary to getting the code/model running; but there are also solver/algorithm choices that depend very much on the physics (e.g., CG is only valid for symmetric problems) - is icepack designed so as to prevent the naive user from inadvertently using the incorrect default setting for their problem? I will assume it is to the extent it can be, and that the authors' argument is that the default hidden settings are likely to do less damage than some arbitrary settings a user might put in his/her input file without knowing what they are doing. Also, I realize that the reality is that many glaciologists do not have strong ties to computational scientists, and still wish to make progress in numerical modeling of land-ice; therefore, I will not focus too much on much "philosophical" perspective described above. Another qualm I have about the paper has to do with performance - I am skeptical whether icepack can really be performant if one tries to run it on continental scale problems, and it is not clear to me if the code is even parallel. I think the intention may be to use icepack as more of a sandbox for prototyping small problems (similar to FeniCS), in which case, this

is not a huge deal. Despite the above concerns, I like this paper and see it published in GMD. I like in particular the idea of describing all the equations using the variational principle/action functional and having everything else propagate from there - not enough people do this. I do, however, feel that there is a lot of missing information in various parts of the paper, which should be filled in in the revision prior to the paper being suitable for publication. Please see below my enumerated list of questions/comments to address in the revision.

**Specific comments**

1. The authors suggest that C++ makes it inherently difficult to add new physics/PDEs (e.g., on p. 10 and p. 20), which I somewhat disagree with. One advancement that can make a C++-based code easy to add to is Automatic Differentiation (AD) - with AD, one can effectively code the weak form of the residual within a C++ code and AD will handle the rest, making it very easy to add new physics. An example is the Albany/Land Ice model (previously known as Albany/FELIX) of Tezaur et al. (https://doi.org/10.5194/gmd-8-1197-2015). I think it would be worth mentioning that there have been efforts like Albany/Land Ice out there to make C++-based codes more accessible to users of varied backgrounds. I agree that even an "easy-to-use" C++ code will be more difficult and more intimidating than a Python code, so I am not trying to minimize the authors' efforts at all.

   I hope this paragraph made the point more that we had specific goals that could better be achieved in Python and not as some blanket condemnation of C++ – I don't like being dogmatic about language choice. I've added a citation to Tezaur et al. and to another paper from the ISSM group about AD. I agree with you about AD tools: they definitely relieve the burden of having to rewrite the code to calculate variational derivatives of functionals of the solutions of the model with respect to input parameters upon changing the model or parameters. The experiences I had rolling my own adjoints are what pushed me to tools that would either have AD or a more symbolic approach like what FEniCS, Firedrake, and Devito offer. In idiomatic Python it's nonetheless possible to be much more flexible about function signatures than in C++ by virtue of being able to throw arbitrary data into `kwargs`. Now of course you pay for this in that all input validation is done at runtime rather than at compile time, but it's a tradeoff we had to make.

2. Can the authors comment on the overhead of the symbolic descriptions/manipulations done by their framework? This sounds potentially like it would be very expensive. How does the cost compare to automatic differentiation, for example? A broader question is: is computational performance/cost a concern for users of icepack, or is it intended to be a "sandbox" in which performance is secondary to being able to code up something "quick-and-dirty" for initial prototyping?

   See Rathgeber et al. 2016 for more information about the architecture of Firedrake and for performance benchmarks. We are constrained only by what

Firedrake can do, so we refer to this paper for performance with no change to the text.

Loosely speaking, the path through the toolchain goes like this. A user creates a symbolic description of the weak form of the PDE they want to solve. Firedrake then computes a hash of this expression and looks to see if it has encountered this problem before. If not, it does a long, complicated, and expensive series of transformations to generate highly optimized C code that fills the relevant matrices and vectors. This C code is then compiled into a dynamic library for later reuse. If the user has solved this problem before, Firedrake simply looks up the dynamic library that it already generated. (Crucially, the only thing that matters is the symbolic shape of the problem, not the actual data that goes into it – you don't have to do codegen all over again just because you changed the boundary conditions or forcing.) In either case, Firedrake then calls into PETSc's scalable nonlinear equation solvers (SNES) to solve the resulting system of equations. As with any just-in-time compiled language, performance is slow the first time the code is run, and faster ever after. Most importantly, **the performance-critical parts are all written in C**. Firedrake has been shown to scale up to large problems on thousands of processors.

It is possible to ruin the performance of the application by accidentally hard-coding a floating point value into a symbolic expression of a PDE and then changing that value in a loop. For example you could easily make this mistake if you were doing adaptive timestepping on an evolutionary problem discretized via the method of lines. The remedy is to wrap this value in a `firedrake.Constant` object. This kind of performance regression is easily caught using `htop`. While this is more a result of programmer error, it's an easy mistake to make and the Firedrake team are working on ways to diagnose it and issue appropriate warnings.

For the problems that we have used icepack for so far, we have focused more on individual glaciers or catchments, and thus performance has been a secondary concern. We aim to move towards larger continental-scale problems in the future. The rate-limiting factor there is more our ability to find the right incantation of PETSc solver options and preconditioners than it is any inherent limitation in our tools.

3. I was a little bit confused about the reference of the FO Stokes-based model in this paper as a "hybrid model". I see that it is hybrid in the sense that you have a different discretization in the horizontal and vertical direction, but there are also hybrid ice models that use different PDEs in different domains, e.g., the ISCAL model of Ahlkrona et al. (https://doi.org/10.1016/j.quascirev.2016.01.032). Is the term "hybrid model" a common name for the approach in Section 2.2.3? Perhaps it is and I am not aware of it. Does the hybrid model described in the paper have the same applicability as say the First Order Stokes model? Can it be used for both Greenland and Antarctica at continental scales?

The fundamental physics are the first-order model obtained by asymptotic expansion of the Stokes equations in the aspect ratio, also known as the Blatter-Pattyn equations. We have amended the text to make this clear. What I had imagined is that using only vertical basis functions up to degree 2 essentially defines its own semi-discrete physics model, similar to two- or three-layer ocean models. You can view these as very coarse discretizations of the primitive equations, or you can view them as simplified models in their own right. But I made this naming choice before there were many other collaborators on the project. The ensuing confusion has shown that this was a bad choice of terminology and we intend to change it in a future version. This model can be used for both regions at continental scales – it can capture both plug and shear flow.

4. Section 2.2.1: in my opinion, the authors do not provide sufficient justification for the penalty term, equation (7). They describe this as something that is added to smooth over artifacts - this would be needed based on the discretization, which there is little discussion of. The authors should state what order finite elements they are using - I presume they are linear, and that this is why the stabilization is needed? Why is stabilization needed only for the SIA? I think these things should be made clear.

See comments by reviewer #3. The technical answer is that this makes the solution live in the Sobolev space $H^1(\Omega)$. A more heuristic answer is that the shallow ice approximation is usually assumed to hold only over distances greater than a few ice thicknesses. The penalty term is meant to filter out variability at length scales where the model doesn't even apply. No change to the text.

5. Section 2.2.2: there is some imprecision here in equation (13) - you have not defined anywhere that $\Gamma$ is the boundary, and which boundary you are referring to. One can figure it out, but it is not precise. $\Omega$ is not defined either though one will assume invariably that this is an open bounded domain in 2D or 3D depending on which approximation one is looking at.

These were not stated explicitly anywhere. We've added them to table 1.

6. The boundary conditions are not discussed very rigorously systematically - the authors seem to sprinkle in some boundary conditions here and there. I think the boundary conditions need to be given for each of the models at the time the models are presented - boundary conditions are needed to complete the definition of each models.

We added this statement to section 2.1: "We implement two types of boundary conditions for the prognostic equation. Users can specify an inflow flux value and this value becomes a source of thickness at any point along the domain boundary where the ice velocity is pointing in to the domain. The flux at the inflow boundary can change in time. Second, we impose outflow boundary conditions on any part of the domain where the ice velocity is pointing outwards. Which segments of the boundary are inflow or outflow are diagnosed

automatically by calculating the sign of the dot product between the velocity and the unit outward normal vector."

We also added an entirely new section which is now §2.3 in the text just on the boundary conditions for the different diagnostic models.

7. Certain terms in the equations I do not believe are defined anywhere, for instance, in equation (10), there is no expression given for the strains $\dot{\epsilon}(u)$. This is one of the things I cam across that need to be made more precise.

This was stated in table 1 but we've added the definition as $\frac{1}{2}(\nabla u + \nabla u^\top)$ and added a sentence to the text referring readers to table 1.

8. Section 2.2.3: the authors comment that higher degree polynomials can be used in the vertical layer in the hybrid approach. What order is typically used?

Added the following text: "Going up to a degree-4 model is sufficient to capture the exact solution for the shallow ice approximation. In the tutorial notebooks for icepack, we use up to degrees 2 and 4, but the test suite checks up to degree 8."

9. Section 2.2.3: this might be a naive question, but does Glen's flow law come into the hybrid model? I was expecting to see it there, but maybe I'm missing something.

This was a bad oversight on our part – the hybrid model does use Glen's flow law and we've added more detail at the end of this section describing the terms in the action functional (equations 22 through 26 in the revised text). The main difference is the term for viscous power dissipation.

10. P. 10: the discussion here about substituting model components suggests it may be possible to use different models in different regions and couple them (a la the ISCAL method). Is this possible, or something that the authors are thinking to add to their model/code?

Implementing this idea will require some new developments to Firedrake, namely first-class support for subdomains, defining different PDEs on different subdomains, and defining matching conditions for the solutions at the interfaces between subdomains. The Firedrake developers are working on this feature right now as it's very much in demand. We are very much interested in, for example, using SIA in the interior of the ice sheet and SSA in the ice streams and margins, as this would give a much less computationally-intensive way to do some form of whole-ice sheet modeling than using, say, the first order model. No change to the text.

11. Section 2.4: This section is very incomplete. You need to give the enthalpy equation and given the Glen's law expression as well, since it is mentioned.

We have added some text and equations describing the model we used, the boundary conditions, and the shear heating rate, the latter of which implicitly includes Glen's flow law.

12. In my opinion, there is not enough discussion of the thickness equation (Section 2.1) and how it is discretized. In typical ice sheet models, this equation is used to change the ice extent - one meshes up a region of "potential" ice, and then uses the thickness to dynamically determine a mask for ice-covered regions. Do you do something like this in your model? It should be discussed for completeness. I think you maybe start to do this in Section 4.1, but it is very confusing and hard to make the connection.

    We added this statement to section 2.1: "Icepack represents the thickness using continuous, piecewise polynomial basis functions in each cell of the mesh. In the examples we use up to degree 2 and the unit tests use up to degree 4. We have not yet implemented a formulation that works with discontinuous basis functions, but this extension is completely feasible within our framework." See also previous comment on boundary conditions for the thickness equation. We also added a longer description of our treatment of ice-free regions (which is very ad hoc for now) at the end of section 2.1. This is a weak point at present and we plan to improve this in future versions.

13. What sort of meshes do you use in your model (in the horizontal dimension, for the hybrid one)? Structured/unstructured? Hex/tet (quad/tri)?

    We use unstructured triangular meshes although Firedrake in principle can use unstructured quad meshes. We've added the word "unstructured" to clarify this.

14. Section 3: It is not clear from the description what the inverse problem you are describing is for. Is it to obtain parameters in the model like the basal friction using observational data of e.g. surface velocity? There really needs to be more discussion here, and I personally would like to see a mathematical statement of a representative inverse problem you are solving. It would be worth citing the work Perego et al. on optimization-based inversion, if what you are doing is similar: https://doi.org/10.1002/2014JF003181. BTW, the basal friction has not been defined, yet it is discussed - it needs to be defined earlier, when talking about boundary conditions (which needs to be added).

    We added a brief description of the mathematics of the inverse problem to be solved. We also added: "The state to be estimated can be any single input field to the diagnostic model – basal friction, rheology, or another field that the user has added by customizing the model."

15. Section 4.1: I find this section confusing. I assume you are talking about discretizing the thickness equation here - that should be made clear. I disagree with several statements in this section as well. "The simplest explicit timestepping schemes are unstable with CG finite elements" - if you are talking about CFL stability, this is not true. You need to satisfy a CFL condition which could give rise to very small time-steps but you can get the scheme to be stable. I'm also confused about the notion of SUPG as a time-stepping scheme - I think of SUPG as a finite element approach to deal with advection-dominated flow problems, for example, that does not have anything to do with time- stepping.

Maybe you are referring to upwinding? In any case, I think SUPG has nothing to do with forward Euler, so the discussion about forward Euler requiring parameters is erroneous. Additionally, I don't understand the comment about implicit Euler smoothing out sharp discontinuities... I believe explicit and implicit Euler have effectively the same diffusion and dispersion properties, so there should not really be a difference between the schemes. Did the authors verify their time-stepper on a manufactured problem to ensure that it was implemented correctly?

This was sloppily written. What we should have said was that SUPG confers some of the benefits of upwind finite difference stencils when using continuous Galerkin basis functions. I was basing the statement about stability on the expression for the numerical amplification factor for the $\theta$-scheme with piecewise linear finite elements from section 3.5.2 in Donea and Huerta, Finite Element Methods for Flow Problems. In any case, we've cut much of this section to focus more on the implementation in icepack rather than what other packages use. The text was also out of date; the actual default now is an implicit scheme with a Lax-Wendroff correction that gives higher order accuracy in time. Our statement that implicit Euler has predominantly diffusive errors was not to imply that explicit Euler doesn't share the same property. We were trying to draw a contrast between what types of errors are tolerable for the prognostic model as opposed to other problems like damage transport, which is described in the next paragraph in the text.

16. Section 4.2: there is an approach discussed in Tezaur et al. (https://doi.org/10.5194/gmd-8-1197-2015) for dealing with bad initial conditions in a Newton solver that relies on homotopy continuation that would be worth citing. It is an alternate to the approach you describe that lets you get away with not doing a line search for Newton. By the way, it should be no surprise that Newton is not converging without a line search - in general Newton is not guaranteed to converge from an arbitrary initial guess without the line search.

Added a reference to the Albany paper as well as another one on trust region methods. We described a fairly rudimentary line search method in more detail than perhaps is necessary for a reader who's a seasoned modeler. It's a bit of a pet peeve of mine when papers just say "We used Newton!" without any attention to the globalization strategy, which can make a huge difference to the solver robustness.

17. Section 4.3, lines 411-412: there are actually ways to construct weighted norms to deal with the issue of DOFs having different orders of magnitude for the purpose of convergence.

I think you can use a lumped mass matrix as the $H_0$ in BFGS too, but I find it to be far preferable to use something like Gauss-Newton which gives mesh-independent convergence and which achieves close to the second-order rate of full Newton on many problems. No change to the text.

18. Section 4.4: Again, it is not clear to me what is your inverse problem. You need to state this explicitly so it is clear.

    See correction to section 3. The point of the inverse solver class is that it is very general with respect to what field is being inferred and what diagnostic model is being used. Since the solver was designed to work for many different inverse problems

19. Section 4.5.1: you talk about problems defined on "extruded geometries" - do you ever use non- extruded geometries? It has been shown in various references that there can be numerical problems for land-ice solvers that do not use extruded geometries, e.g. Tezaur et al.

    We have restricted our implementation to use only extruded geometries. Having made this choice, we might not ever be able to solve really geometrically complex problems like what the Elmer/Ice crowd did with the drainage of the lake underneath Tête Rousse glacier. But the simplifications that this results in for the vast majority of glaciological applications are so advantageous that it's a sacrifice we're willing to accept. No change to the text.

20. Section 4.5.2: I think this section needs to be made earlier, and other BCs need to be added to that discussion.

    See response to previous comment and the additional section we added on boundary conditions. This section describes some extra care that we had to do in our implementation which was purely a consequence of our choice of basis functions and which does not appear in the idealized mathematical form of the model. In keeping with our overall goal of splitting the paper up into a section on what we're solving and a different section on how we're solving it, we've kept this section where it is.

21. Section 4.6: The authors mention running their code on 1 core. Is the code parallel - can it be run on multiple cores? Are there any hope for performance portability of the code to take advantage of emerging HPC architectures, e.g., GPUs?

    Firedrake relies on the package loo.py for code generation, which can target C and OpenCL. There is ongoing work with the developers of loo.py to target GPUs and other accelerators by generating OpenCL instead. Firedrake is built on PETSc and thus can run on parallel machines. See again Rathgeber et al. 2016 for performance and scaling benchmarks; Firedrake has been run on problems with millions of degrees of freedom on supercomputers, for example the UK national supercomputer ARCHER. No change to the text.

22. Section 4.6: Can you please clarify what you mean by the following statement? "Large problems, such as continental-scale modeling, will require more sophisticated and possibly problem-specific approaches". I'm wondering in particular about the problem-specific approach part. There are models like first order Stokes that can be used at the continental scale and they are not really problem specific.

We have clarified the text to state that "problem-specific" refers more to the strategies we use to solve the resulting nonlinear systems of equations rather than to what equations are being solved, e.g. Stokes vs first-order Stokes: "For the demonstrations presented below, nearly all simulations run in a matter of minutes to hours on a single core. We have used sparse LU factorization to solve linear systems for many problem instances in order to eliminate the linear solver as a possible failure mode. Defaulting to a robust solution method is especially important for onboarding novice users who may not be familiar with different iterative linear solvers and preconditioners. Larger problems, such as continental-scale modeling, will require solving the diagnostic equations using the conjugate gradient method with an appropriate preconditioner to achieve parallel scalability. The particular structure of the problems we solve may be useful in choosing a preconditioner. For example, a rudimentary preconditioner for the hybrid model system could use the degree-0 model as the coarse space in a multigrid-type approach. These optimizations will be the subject of future work."

Since we use a modal basis in the vertical to discretize solutions of the 3D model, we can devise a p-type multigrid scheme along this axis. This is an example of using problem-specific knowledge to choose a solution strategy. Just using LU or throwing a black-box algebraic multigrid preconditioner at it would be failing to use this special structure. That said, we did not want to speculate too much on approaches that we haven't implemented yet.

23. p. 20: I don't understand why you need to create an analytical expression of (32) using special functions. Is this something specific to your framework, which requires expressions in a certain form for the symbolic representation?

We don't need an analytical expression of equation 32, but rather of the antiderivative of that function with respect to $u$. This problem is specific to icepack because we have made the choice to use action principles to describe all of the diagnostic models. A package that also made the choice to use action principles but which was built on a different finite element modeling library or coded in an entirely different language would also need the antiderivative of this function. We are hampered by the fact UFL does not include support for hypergeometric functions. If we were instead writing everything from scratch in C++, we could call into Boost or GSL to evaluate hypergeometric functions. We believe that the benefits outweight the costs but this is a definite drawback of our approach. No change to the text.

24. It's great that you have executable documentation in something easy-to-use such as Jupyter notebooks! (no need to address this comment)

25. p. 29: are you considering putting in the first-order Stokes/Blatter Pattyn model into your code framework?

The thing that we mistakenly called the "hybrid" model is really the first order / Blatter-Pattyn model. We have rewritten some of the text to try and make this clearer.

26. The methods do not discuss their code development/testing stance on icepack. How do users contribute to the code - through pull requests? Is there regression/performance testing? Continuous integration testing? These are all really important, especially if you have non-experts contributing to the code!

This information is on the icepack website (https://icepack.github.io/developers). Users contribute through pull requests which are automatically checked against a regression testing suite. We try to keep the test coverage at 95% or higher. There is at present no automated performance testing short of looking at the timings from our CI service. Our development practices have changed appreciably even during the process of writing this manuscript; several of the contributors are students who are learning more about version control in tandem with learning to implement new or modify existing models. We have added references in the code and data availability section about where to find this information.

**Minor comments**

- "UFL" is not defined in the abstract.

    Changed the sentence to: "Icepack is built on the finite element modeling library Firedrake, which uses the Unified Form Language (UFL), a domain-specific language embedded into Python for describing weak forms of partial differential equations."

- p. 4, line 90: A is also called the "flow factor". I would mention here that it is usually a function of the temperature, which comes from a different equation.

- p. 5, line 125: should be "checking", not "check". ✓

- p. 5, line 126: I think it should be "Bueler" not "Beuler", if I'm thinking of the right person. ✓

- p. 5, line 147: change "we verified the correctness of the ice shelf model" to "we verified the correctness of our implementation of the ice shelf model". ✓

- p. 10, line 270: change "we'll" to "we will". ✓

---

## Author Comment (AC3)

**Response to review #3**

Thank you for this thorough review! Original comments are in blue, our responses are in black, and hyperlinks in magenta.

**Summary**

Icepack is an important new model, and a description paper is appropriate. Four features of Icepack stand out: its use of Firedrake/Python, the flexible action-principle design of its stress-balance solver module, its from-the-start attention to data assimilation and inverse modeling, and its design as a modeling environment and language instead of a ready-to-run model. All of these choices are addressed appropriately in this manuscript. Readers are very likely to try the model, which fullfills a major purpose of a description paper at this early-ish development stage.

However, this description paper can be improved in three significant ways:

A. Greater attention to the meaning of Icepack as a *time-dependent and geometry-evolving model*, and to related defaults.

B. Avoidance of *bad linguistic habits* inherited (mostly) from the ice sheet modeling literature.

C. As currently laid out, the paper treats the reader mostly as a potential co-developer of the model, while Icepack's *effectiveness as a simulation tool for science* is muddled.

Fully addressing concern A would require major code extensions, which are not my intention in commenting on this. Rather, the manuscript should make the reader aware of which evolution aspects are well-handled by current Icepack and which are in future development.

The above leading concerns will be addressed in more detail below, in a list which addresses specific line numbers. The associated potential improvement "A","B","C" is listed when appropriate.

**Line-by-line comments**

13, AC: Presumably all software packages can be effective in the hands of experts, and here it is suggested that Icepack stands-out as better for non-experts. My question is whether it will be effective for non-experts interested in announced goals 1 and 4 (among those listed on lines 15–18). This is hard to believe given how Icepack seems to be designed around, and the manuscript focused on, goals 2 and

3. Consider the reader who wants to simulate glacier extent for some years into the future for a mountain glacier or Greenland, who can supply a simulated climate (atmosphere/ocean), a bed topography, and a current geometry in some data files. Does this paper convince me that they need *less* expert knowledge to use Icepack for that purpose than other existing ice sheet models? Not yet. Of course a full usage answer occurs in online tutorials and examples, not just the manuscript. Nonetheless the absence of attention to transport equation boundary conditions, and to how mass/energy surface inputs are handled, gives me the impression that such a reader is on a long co-development path with the Icepack authors, requiring the development of much expert knowledge before first useful results. Said a different way, expert knowledge is required for any software that does not have an aggressive scheme for putting reasonable defaults into the hands of novices. To paraphrase a recent Firedrake paper [Farrell et al 2020], it is a mechanism-vs-policy concern. Icepack has a library of mechanisms, but an absence of apparent policy means expert knowledge is needed to recover usability for real-science applications. I rolled my eyes at the implications of the sentence on line 13, and these concerns remained after reading the whole paper.

This depends on what kind of expertise you mean. We believe that more work can be done to relieve glaciologists *who are not experts in scientific computing* from having to understand, e.g., the details of what preconditioners are used in solving linear systems, what globalization strategy is used in a Newton solver, etc. By way of an "aggressive scheme for putting reasonable defaults into the hands of novices", we describe our policy at the end of section 6, beginning in line 695. Other software packages have (exactly as you say) all the mechanisms, but the policies on display suggest the priority is for speed above all else. For example, several of the demonstration codes in Elmer default to choices of iterative solvers and preconditioners that work on the particular problem in the example but which are likely to require hand-tuning when generalized to real data. See for example this line from Elmer's demonstration of an inverse solver, which uses the GCR scheme and an ILU0 preconditioner to solve the momentum equation. A graduate student could easily try to take this code and use it on real data only to have convergence failures to diagnose. Other examples from Elmer do use direct solvers, for example their demonstration code to run the ISMIP test case with the SSA model uses a direct solver from UMFPACK for the momentum equation. Other packages show a similar pattern – BISICLES has hand-tuned numbers of Picard iterations and multigrid smoothing iterations in their demonstration codes. A new user either must (1) know enough about numerical methods to diagnose the problem from bad input data, bad linear solver options, or bad nonlinear solver options, or (2) have a mentor who can. Experienced users have all the tools at hand that they need to create large simulations that will run on supercomputers, but at the same time these choices can create friction for people who may be getting their first experience of modeling by using these examples.

We do not claim to have reduced the need for expertise in glaciology itself, i.e. understanding mass and energy balance, etc. We also consider having some understanding of the different boundary conditions that might apply to the system to be part of this knowledge. The text has been expanded to describe these boundary

conditions in more detail (see comments that follow). Granted, (1) there is more demonstration work that we need to do in order to cover use cases like modeling a small mountain glacier and (2) there are technical hurdles we still need to overcome to do a better job handling the margins. Very likely your recent work on variational inequalities will factor into improvement on the second front.

We have altered the text to state more clearly that our goal is to help users who are not experts in scientific computing. We added the following to the introduction: "We have focused efforts thus far on process studies of individual glaciers or drainage basins (use cases 2 and 3 of the list above). Development of icepack is ongoing and we will broaden our efforts to encompass more use cases in future." We also added the following sentence to the conclusion: "Exactly how to apply principles from HCI to maximize usability is, nonetheless, not an exact science. Our implementation may have failed to meet this goal and changes in future versions will be guided by what users find most difficult."

21–39: I also think Python+Firedrake+PETSc is the most promising environment to build a new ice flow simulation library/model. I'm on board!

44–46, B: Let me vote to *not* maintain the "diagnostic"/"prognostic" linguistic tribalism. The world calls these equations "conservation of momentum" and "conservation of mass". (Indeed one should remind the reader that the latter describes thickness evolution because of how glacier models normally parameterize fluid geometry.) An "also known as the 'prognostic equation' [cite]" is appropriate, but another paper using this tribal language will cause yet more students to need to unlearn silly language in order to read the mainstream fluids and numerical PDE literature.

I agree with you that this is not great terminology, but this isn't a fight we could win. Many other software packages for glacier flow modeling, including PISM and ISSM, use this terminology. No change to the text.

46–47: Here! Here! It is a infinite-dimensional DAE! Good. Many readers will be unfamiliar with the concept; I cite [Ascher & Petzold 1998] for that but there may be better references.

Added a reference to Ascher and Petzold.

56, equation (1) and nearby, AB: Two concerns. First, it is later acknowledged that this is not really an advection equation (lines 376–377), so one does not need to call it that here either. The SIA is not some weird alternative universe of weak-willed modelers, it is what *all models should produce* in the large. That model, the only clearly-understood coupled model, makes equation (1) a diffusion, as noted. Surely calling this a "transport equation" or even "thickness transport equation" is adequate. One then points-out that $q = hu$ is one way to parameterize flux, and that $u$ comes from a coupled equation *driven by* $\nabla s$ (even in the Stokes case). It might be acknowledged that (1) does not have a PDE "type" in the classical sense. (It is a DAE, after all.) Second, equation (1) holds with what boundary conditions? This manuscript maintains the tradition of pretending not to notice. To quote [Schoof & Hewitt 2013], "A sometimes weakly perceived point in glaciology is that the model above is in fact a free-boundary problem ...", and *this applies to (1) regardless of the stress balance model*. Is this paper just going to pretend boundary conditions for the main, and first, equation don't exist? (Or pretend that all glaciers end in cliffs

of sufficient height so that the singular change of coordinates (16) is no problem?) This manuscript could even cite existing wellposedness literature for (1), especially [Calvo et al 2003] and [Jouvet & Bueler 2012], which confirm the meaningfulness of the free-boundary problem formulation and the resultant simultaneous Dirichlet (h=0) and Neumann (q=0) boundary conditions at grounded margins. And then say what Icepack will when solving (1).

We removed the statement that this is an advection equation and added the following sentences: "This problem has the apparent form of a conservative advection equation, but the velocity $u$ is coupled to the thickness and surface slope in such a way that the whole problem is not hyperbolic. For the specific case of the shallow ice approximation (see section §2.2.1), the coupled system is parabolic. In all other cases, the problem does not have a PDE 'type' in the usual sense because the velocity is found through solving an elliptic PDE where the thickness and surface slope are coefficients."

Our discussion of boundary conditions was mostly lacking. We added the following:

"We implement two types of boundary conditions for the prognostic equation. Users can specify an inflow flux value and this value becomes a source of thickness at any point along the domain boundary where the ice velocity is pointing in to the domain. The flux at the inflow boundary can change in time. Second, we impose outflow boundary conditions on any part of the domain where the ice velocity is pointing outwards. Which segments of the boundary are inflow or outflow are diagnosed automatically by calculating the sign of the dot product between the velocity and the unit outward normal vector.

The mass transport equation for ice thickness is a free boundary problem, where the free boundary is the contour between ice-covered and ice-free regions (Schoof and Hewitt 2013). A naively-implemented prognostic solver could erroneously compute negative thickness values in subsequent timesteps when there is ablation in ice-free regions. A common and ad-hoc approach to work around this issue is to truncate the thickness at zero at every timestep. The principled approach is to instead treat the free boundary problem directly as a variational inequality (Jouvet and Bueler 2012). Icepack currently lacks a principled scheme for tracking this free boundary and we instead rely on truncation. Treating this problem as a variational inequality in icepack will be the subject of future development. PETSc includes scalable solvers for variational inequalities (Bueler 2020) that are also available through Firedrake."

We also added a new section which is now section §2.3 in the text on the boundary conditions for the diagnostic models.

68, B: It is not clear why the conservation of momentum model for a slow fluid should ever be called a "momentum transport" model. There is no d/dt for momentum! Surely "conservation of momentum equation" or "stress balance equation" suffice.

Changed from "momentum transport" to "stress balance".

74, B: No, the Blatter-Pattyn (BP) equation is not a "hybrid". This word is perfectly descriptive when categorizing ice sheet momentum models, and I am not sure who decided to mangle the language in this way. (It precedes this manuscript.)

There are principled models, of which SIA, SSA, and BP are three, arising from small parameter arguments. There were horizontal hybrids (Ritz) and there still are vertical hybrids (Pollard, Bueler, Winkelmann). Hybrids, by definition, combine more than one principled model in some (less principled!) manner. If you are trying to say that BP is a shallow model, but less-shallow than SIA and SSA ... that's true! And if you are saying that BP can balance stresses in a plug or a shear flow, that's true too. Say those things.

This was bad terminology. I adopted it before there were many other contributors to the project and everyone who has come along afterwards has told me that it's confusing. What I had imagined is that a discretization of BP using only a single vertical layer and relatively low-degree (up to 4) polynomials in the vertical direction could be considered a kind of approximate model in its own right, similar to how oceanographers will use semi-discrete two- or three-layer ocean models. In the next release we will change the name to "Blatter-Pattyn" or "First Order" and deprecate the name "Hybrid" for removal in a future release or until we develop an actual hybrid model which is less computationally intensive than this Thing That I Should Not Have Called a Hybrid Model.

101–102, B: Regarding the sentence "The action principle can be viewed as a consequence of the Onsager reciprocity relations for systems near to equilibrium ...": I am trying to imagine a reader for whom reading that would be a useful learning experience regarding the action principle, as opposed to an irritating one.

The previous two sentences have been removed from the text but we've kept the reference to the book by De Groot and Mazur. When I've spoken to students about action principles and why they're useful, I usually highlight the numerical advantages of solving convex optimization problems as opposed to the more general problem of solving large nonlinear systems of equations. But the question invariably comes up as to why a given physics problem should have a minimization principle at all while others do not. Non-equilibrium thermodynamics provides some insight and I think we would be doing a disservice by not mentioning this at all.

108: I do not understand the intended meaning of "vastly more convenient numerically" here. The cost of evaluating the objective, versus a merit function like the square of the residual norm, is the same. The cost of evaluating a residual is the same whether or not it happens to be a gradient of an objective. Perhaps what is meant is more like: "Minimizing a convex functional provides better convergence guarantees than solving a general system of nonlinear equations ...", which is certainly true. At least a forward reference to section 4.2 would help explain the phrase.

Added a forward reference to section 4.2 and changed this sentence to "Algorithms for minimizing convex functionals have better convergence guarantees than algorithms for solving general nonlinear systems of equations while having no additional computational cost" with a reference to Nocedal and Wright.

115, equation (5), B: Now the bad linguistic habits are being inherited from the FE world. This is not "mass" in any sense other than that any matrix with entries $a_i j = \int psi^i psi^j dx$ is always called a "mass matrix" in the FE world. And the reader may not have that FE bad habit already, in which case confusion ensues because (5) has nothing to do with mass conservation! My suggestion for this term, implicitly

acknowledging how it arises in the SIA, is to call it "localization".

Changed to "localization".

115–124: The action combining (5) and (6) is already convex and coercive *in $L^2$*. The reason to add the penalty term is (presumably) because the FE space choices want to work in $H^1$. (Adding (7) with $\ell > 0$ makes $J$ coercive in $H^1$.) This penalized form is a perfectly reasonable idea, and it makes the SIA behave more like other stress balances, *and* it is one of those unprincipled things one does to get it all to work properly ... if you add another kludge you could even call yourself a "hybrid".

126: This author's name is Bueler not Beuler. (The latter is a PETSc `-ts_type`, so it is easy to get confused.)

Apologies, the text has been corrected.

137–140: There is a double negative in equations (12) and (14) which is not used in (6) and (8), respectively. Does this reflect anything important?

These modules were developed by different authors. We will get rid of the discrepancy in a future version.

148–149: It is of course true that for general boundary conditions the "shallow stream equations do not have a simple analytical solution." What is probably meant here, however, is that there is not a well-known, exactly-solvable, basal-friction-included boundary value problem suitable for testing. But that's not true, and indeed the *very first theoretical paper in glaciology* provides one, namely [Böðvarsson 1955]! See the full story, and a derived marine flow-line exact solution suitable for testing, in [Bueler 2014]. The method of manufactured solutions applies, of course, but at loss of clarity on the meaning and magnitude of the resulting numerical errors, and with great danger of testing the wrong parts of a (nonlinear) system phase space, and with loss of the history of mathematical glaciology.

Added a reference to Bueler 2014 and Böðvarsson 1955. You make a great point about testing the wrong parts of a nonlinear phase space and this was actually a bit of a struggle when creating the tests for this solver. You could put in some thickness and velocity fields and a PDE and SymPy will spit out a manufactured friction, but there's no guarantee that the result will be physically reasonable. I had to manually adjust the input parameters so that the basal shear stress values would come out to something sensible. A sentence has been added to the text to describe this part of the testing process.

158: I guess the phrase "individual fast-flowing glacier" arises because of the sense that the SSA does not handle slow flow very well, which is true, and that fast ice is separated by slow flow. Nonetheless, I would replace "features of an individual fast-flowing glacier" by "features of fast-flowing glaciers".

Done.

170–171: I am not clear on "a tensor product basis of Lagrange finite elements in the vertical and higher-order polynomials in a single vertical layer". Should the first "vertical" actually be "horizontal"? The novice reader not already used to FE stuff could use a figure or more words here, I suspect.

That was a typo, fixed.

175, equation (16), A: From here forward, in the manuscript, the lateral boundary condition for equation (1) is even less clear because this change of variables

takes the lateral free margin and blows it up. (In the grounded ice case, and where "blow up" is used in the mathematical sense.) With what consequences for simulating moving margins? Section 4.1 does not address this (i.e. how the boundary condition for the transport equation behaves as h->0; a boundary condition is not mentioned). I believe, on the other hand, that the SIA grounded-margin example in section 5.2 does not use (16), which as described applies only applying to "the hybrid flow model", but section 5.2 does not mention the margin anyway. I am concerned that positive consequences of using (16) are emphasized while negative consequences are not mentioned, and that Icepack has no path around these negative consequences.

Section 2.3 in the text describes the lateral boundary conditions for the diagnostic models and an admission that we assume the entire spatial domain is ice-covered. It's possible that I've painted us into a corner with this choice. In the $h \to 0$ limit, the only singularity in the equations is in the vertical shear stress, which – including a factor of $h$ from the determinant of the coordinate transformation – scales like $h^{-1}$. A dirty hack would be to replace this term with `1 / max_value(h_min, h)`, effectively clamping it at the reciprocal of this minimum thickness. This would still permit ice-free areas at the risk of underestimating vertical shear stresses in areas where the ice is thinner than whatever threshold value was used. Alternatively, we may have no choice but to switch to a $z$-coordinate model in future versions.

215: For clarity to readers not already thinking about Firedrake's extruded meshes, I suggest adding a parenthetical: "linear or quadratic elements on (horizontal) triangles".

Done.

216: Instead of "Lagrange interpolating polynomial basis", which readers accustomed to numerical analysis terminology (outside FE) might associate to the Lagrange method of finding interpolating polynomials, it is probably clearer to just say "Lagrange elements". Perhaps emphasize the nodal placement? (Legendre nodes, concentrated near endpoints, versus the equally-spaced nodes of Lagrange elements.) Orthogonality is not the only reason why Legendre is better; node location implies better approximation properties.

Done and added a reference to Szabó et al 2004.

227: "Smoke test" is an unfamiliar idiom to me. As in testing machinery or testing electronics or smoking-out bugs or what-am-I-smoking? Pity the non-native speaker.

The term comes from electronics. "Plug the circuit board in. If you see smoke coming out, unplug it. You don't need to do any more testing." Changed to "sanity test".

227: "most minimalistic" –> "minimal"

Done.

234–235: I have received the advice "write about what you have done, not what you haven't", and I think it applies here. Using two layers in this way assumes one polythermal structure and will not allow transition to the other. No need to consider or mention it.

These sentences have been removed.

253–289: This chunk of text is important, and it describes one of the best aspects of Icepack's use of Python. I think it should be put at the start of subsection 2.3, which will give clarity that when equations (22) and (23) appear, they are primarily an example of submodel plug-ability.

283–289: I'm on board with keyword arguments!

298–299: Suggested replacement for the mushy phrase: "describing realistic glacier flows, but it is not sufficient by itself" –> "parameterizing fluidity".

Done.

300–301, C: I seriously doubt that more than one user in 100 will want to substitute their own melt-fraction dependence, because no one has the data for it. So: "likely" –> "possible". (This is a clear case where good defaults are more important than modularity.)

Changed to "Some users of this module may want to substitute in their own parameterization for melt fraction dependence."

297–305, C: This paragraph reads, to me, like an argument that expert knowledge *will* be required for the Icepack user. ("users must calculate the fluidity field themselves") That is, unless Icepack includes both defaults and higher-level simulation drivers which allow the novice user to never know about these possibilities. More broadly, what does Icepack design offer, when it comes to avoiding explicit choices of all submodels, on the way to a first scientifically-useful model for the beginner?

What you're describing is knowledge of glaciology, not scientific computing. We have stated explicitly in the first sentence of section 6 that we aim to reduce the need for users to be experts in the latter but not the former. If we were to define how some fields are coupled, we would be faced with a difficult design choice of where this is or is not appropriate. Including the effect of temperature on rheology seems fairly obvious. Some modeling papers also account for much higher fluidity for ice that accumulated during the last glacial period because of the greater dust content. Should it also be the responsibility of the package to include this effect? We did not feel that, at this stage of the development, we could also implement high-level simulation drivers that hide these details while also satisfying our other design goals. See also response to the first comment and the text added to the conclusion. We believe that our design choices will ultimately enhance usability for the intended audience and we did our best by applying principles from HCI. But we may well have missed the mark and the interface may change to respond to evolving understanding of our target audience's needs.

318–320: Mushy sentences. Needed?

We wanted to emphasize that this model is very ad hoc and that we're not passing it off as more accurate than it is. These sentences have been removed.

345, C: The sentence "The key classes that users interact with are flow models and solvers" is right at the heart of my concern C. Indeed, after developing PISM with much (developer group) concern about flow models and solvers, we spent the next 10 years talking to users about their data formats, parameter-study schema, and surface process models (supra, sub, and calving front). Barely had time to work on flow models and solvers ... until I quit day-to-day PISM development. The by-far largest population of scientists care about how flowing ice interacts with their

climates. They use ice flow models on the assumption that ice flow modelers know how to model ice flow! This "key classes" sentence is not describing users, it is describing co-developers.

Again I think this seems to be a disagreement about what the software package should do and what the users should be responsible for. Icepack lacks any of the high-level simulation drivers that PISM has. Giving users a relatively lower level interface to the physics solvers was a concession to the fact that we can't predict what they will want to do with an ice flow model by way of coupling to climate, oceans, solid earth, etc. Obviously you and the developers of PISM have had far more experience. We may well have misjudged what glaciologists do and don't find difficult about modeling and what type of interface best serves the needs of the most people. See response to first comment.

368: "Practicing ... may be unfamiliar with" is unnecessary. Just say what you have to say.

Changed to "Some glaciologists may be unfamiliar with the Courant-Friedrichs-Lewy (CFL) condition."

362–381, A: Three concerns about section 4.1. First, as already noted, transport equation (1) has boundary values and they are again silent here. Second, there are multiple transport problems in glacier modeling: equation (1), the advection-dominated enthalpy equation, the ice shelf damage transport equation, the age equation, ... etc. Is this paragraph covering them all, or just equation (1)?

See response to previous comment on boundary conditions. We cut some of the text in this section in response to comments from reviewer #2 and clarified that we were referring to the mass transport equation.

368–373, AC: The third concern, regarding this paragraph, is that Icepack apparently has not yet set-up effective adaptive time-stepping. (I have not gotten a runtime error from a modern explicit ODE solver for a long time! Have you? Some runs are slow cause they are stiff, indeed.) This should be addressed/acknowledged. In particular, assuming a perfectly-implemented *implicit* solver, how does $\Delta t$ scale with $\Delta x$? If the problem were really advective, so that the goal is to model influences as they travel at the characteristic speed, then the answer would be $\Delta t = O(\Delta x)$ for accuracy, even with your implicit scheme. (Better than $O(\Delta x^2)$, yes indeed ... but then explicit steppers would be fine ... but (1) is not an advection ... we must think more.) Though the problem is actually some diffusive/advective mix, some defined scaling of $\Delta t$ with $\Delta x$ is still needed for accuracy. This can come from an adaptive time-stepper, or be designed from scratch. Then there is the matter of the user's data time scale for surface mass balance, etc., which implies "events" in the adaptive time-stepper (when data is read). In fact, the actual questions a glacier model must answer, regarding time steps, are these: How does time-stepping change under spatial grid refinement? Does it converge in space-time refinement? Is it robust over realistic geometries and inputs? What is the user interface? This paragraph convinces me that Icepack is not yet there, which should not be completely buried.

Text has been reorganized, and we added "Icepack currently lacks an adaptive timestepping scheme. Unconditionally stable schemes allow taking long timesteps, but taking very long timesteps will give inaccurate solutions. At present, users are

still responsible for checking the accuracy of their results, for example by running at more than one resolution. Adaptive timestepping will be added in a future release."

This is one of the areas where we don't (yet) live up to our goal of keeping glaciologists from having to worry about numerics. Our plan is to use the package firedrake_ts, which offers an interface to the PETSc timestepping schemes.

363: "equation equation"

Fixed.

365, A: The "In the interest of simplicity" phrase here tells me the authors simply have not thought-through the time-evolution of glacier geometry and the needed boundary conditions. (You have a scheme for unconditional-stably generating glacier surfaces which conserves mass in the presence of ablation? Then don't hide it!) Please take the problem seriously: Address how you maintain reasonable margin shape and positive thickness in an implicit scheme. It is o.k. to admit that time steps cannot be arbitrarily long in your scheme, to get convergence; the "conditional/unconditional stable" language is an artifact of linear PDE theory, and your problem, taken seriously, is super-duper-nonlinear.

See response to previous comment. We've changed the text to say that the scheme is unconditionally stable for the advection equation with a note that the coupled system is not linear nor is it hyperbolic. We also changed a later sentence to: "Implicit schemes allow taking longer timesteps than explicit ones, but taking very long timesteps will give inaccurate solutions and, in the presence of ablation, may yield negative thickness values."

376–377, B: This sentence is a very good argument for *not* calling (1) an advection equation.

383–444, A: Sections 4.2 and 4.3 convince me that when the Icepack team takes on aspects of model design they care about then it comes out very well! These sections suggest the momentum balance ("diagnostic" ... grrr) solver has great defaults. Likewise with sections 4.4 and 4.5. (Now for serious attention to time-stepping, mass conservation, and the user's surface mass- and energy-flux data, and clarity on Icepack's TODO list.)

459–462: Gauss-Newton is a good choice for this purpose, as pointed out for exactly this purpose, inversion of glacier models, by [Habermann et al 2012]. Which should be cited.

Added a citation to Pratt et al 1998 and Habermann et al 2012.

463–466: Symbolic derivatives. Again I am on board with the benefits of a Firedrake-based tool chain.

469, A: Here "terminus boundary condition" refers to the momentum balance equation. There is, as far as I can tell, no regard for the "terminus boundary condition" needed when solving equation (1), i.e. the mass conservation terminus boundary condition, even in the case of a cliff, much less a grounded margin.

See response to previous comments.

474–487: Extruded meshes. On board with Firedrake. But if the model claims to have a solution of (1) then there would be some mechanism for addressing regions which become ice-free within a time step? No mention thereof.

You can use extruded meshes of variable thickness and that thickness can become 0. So the choice of extruded meshes does not force one to use terrainfollowing coordinates as we did, whether or not that was a good idea. No change to the text.

493–515, A: Section 4.5 describes a nice solution to a real problem. In fact, can one consider this kind of fit to the glacier profile at a grounded margin? Not obvious how to proceed, I agree, but note that the thickness field h of a glacier has the similar continuous-but-non-differentiable character as the pressure on the ocean calving front.

The technique might be applicable in other areas but we haven't had the chance to try it yet. No change to the text.

517–522: Rich problem-aware preconditioners and solvers. On board with Firedrake.

523, A: This honest, clear sentence is not at all expected by the reader who remembers lines 15-19 in the Introduction. Please let the reader know earlier.

We added the following to the introduction: "We have focused efforts thus far on process studies of individual glaciers or drainage basins (use cases 2 and 3 of the list above). Development of icepack is ongoing and we will broaden our efforts to encompass more use cases in future."

533–586: The MISMIP+ experiment is part of the 2008-onward tradition of running ice sheet models in rectangular boxes (MISMIP, EISMINT-HOM, ...). This is fine in a field where one also has some laboratory fluid models, i.e. actual stuff, which fit in boxes. But this tradition has a distressing impact on new model development, which is to modularize around the choices one makes between intercomparison-specified boxes and their associated boundary conditions. (Yes, that is what Icepack looks like right now.)

We included this assuming that it's a necessity to repeat at least some of the standard benchmarks from the literature in order to publish a new model. No change to the text.

588–597, A: This example would be more impressive with hybrid physics, right?, but Icepack is not there yet. (Or else it would have been applied here.) Readers of GMD should be taken to be serious people. Tell them the score, and I don't mean relative to what other models are capable of. What needs to improve?

See response to previous comments. The point of the exercise was to reproduce at least some of the results from a prior paper that used the shallow ice model exclusively.

598: Regarding "computationally cheap enough ... in a matter of minutes on a desktop": I suppose the excuse for this sentence is that other people get away with writing such stuff? The question is how run time scales with mesh resolution. The reader can handle a plot, if you want to generate one.

We avoided discussing performance or scalability considerations in this paper. See our responses to reviewer #2. These will be the subject of future work.

604–647: Nice examples! Ice shelf physics is a scope where Icepack is a convincing choice for a research project.

662–663: Regarding "The person learning ... is largely absent from the discussion", have you looked at the PISM User's Manual? (Start from page 1.) This opinion ranges from disputable to insulting, but mostly reflects not paying too much attention to unpublished prior literature (i.e. online manuals). On exactly that topic, the

online Icepack documentation is excellent.

This sentence has been removed.

666–713: Saying these HCI principles out loud is a very worthwhile aspect of the manuscript.

677–678, C: There is a \*big\* gap between "user interface in an interpreted language" and a "program that can only run in batch mode". C-based PETSc programs like the PISM ice sheet model, and many other command-line programs, live in the interior of this gap. For example, please consider these three HCI principles as they apply to command-line git. It is neither "a user interface in an interpreted program" nor a "batch mode" only thing. (Most command-line tools are not!) Git has a steep learning curve, because it is a DAG modeling language, but such command-line design can hit your principles too. Indeed, a good antidote to your false dichotomy is [Brown et al 2014], and addressing UI points made there would increase the credibility of section 6. (You'll see that Icepack is definitely a Brown-approved design library-wise, but doing science is not equal to designing a library API either.) Ultimately any science application of Icepack will be a map from inputs (data) to outputs (simulations or inversions), and the usability of that map (e.g. in ensemble simulations) is different from the develop-ability of it in an interactive environment. Python is a great environment for experimentation, and for ice sheet modeling, and progressive evaluation is a desirable principle, but an interactive Python session is not the only alternative to 1990s climate-model design. Presumably Icepack usage is intended to progress from all-interactive mode to parallel production/ensemble runs in batch systems anyway? Address that?

We added the statement: "Simulations that have been debugged can then easily be transformed into a single Python script, for example using the tool `nbconvert`, for production runs and parallel execution. In other words, while there is an interactive interface, there is also a faster batch mode interface as well."

681–683, C: Regarding "The API ... one step at a time": There is a \*huge\* amount of expert knowledge required to do science with an "ice sheet model" which does not have a policy for doing things "one step at a time". Imagine a paper that proposed a new-and-better WRF model but said something like this about the demands on the user!

682–683, C: Also, this "user" is a co-developer.

Several authors have noted the necessity of providing a path for users to become developers for the long-term health of an open-source software project. See for example Turk (2013), Scaling a code in the human dimension or Bangerth and Heister (2013), What makes computational open source software libraries successful?

693–713, C: This basic point about abstraction gradient is good. But solver/discretization components/choices is not the only such gradient. From talking to a lot of novice glacier modelers, I assert the key abstraction gradient, which an ice sheet model must finesse, regards ice flow physics. (What aspects of the full, coupled, nonlinear dynamics are the next ones that the user's constructed Icepack model should/can handle for the science goal? How to build-in that physics without unnecessary parameters? How to generate intermediate results which reveal which processes are missing, re the science goal?) Thinking on this stuff is where Icepack developers could make their next real progress. That means de-emphasizing solver components/discretizations in the user's view, primarily by setting aggressive defaults, even as the developers must get solvers right (which is the strength of Icepack).

It's possible that icepack will evolve to have a more end-to-end simulation driver as the main interface. See changes to intro and conclusion.

721: "phyics"

Fixed.

724: Having reviewed quite a few ice sheet modeling papers, every single one claimed something about its usability. Less mush, more answers please.

Changed to "Relatively few works in the computational science literature draw directly from relevant work in HCI when discussing usability; see for example Hannay et al. 2009, Harris et al. 2020."

725: I'm on board with (2), and wish for more careful analysis of Icepack's contribution here, but "quantifying" in (1) is not what you are doing in this manuscript. (Nor, probably, do you have the ability to do it.)

Expanded to: "We believe that this is because of two difficulties. First, the degree to which usability is a rate-limiting factor for scientists is hard to quantify and likely differs widely across disciplines. Second, concretely assessing what features make software tools more or less usable is highly subjective." We're making an unsubstantiated claim that usability is indeed a rate-limiting factor. We could be wrong about this. Even if we're right, the steps that we took to address this problem could have been misguided.

732–740, AC: If I saw "4. robust time-evolution and climate interaction tools", or similar, on this list then I would be more of a believer in Icepack's future. Will you be able to break out of stress-balance-and-inverse-model-solver-playground mode, and start answering some of the science questions enumerated in the Introduction?

We have changed the list to include "improved physics formulations and solvers that work in ice-free areas".

**References**

Ascher, U. and Petzold, L. (1998). Computer Methods for Ordinary Differential Equations and Differential-Algebraic Equations, SIAM, Philadelphia.

Böðvarsson G (1955) On the flow of ice-sheets and glaciers. Jökull,

5, 1–8 (find this too-little-read paper at: https://github.com/bueler/bod-marine/blob/master/Bodvardsson1955

Brown, J., Knepley, M. G., & Smith, B. F. (2014). Run-time extensibility and librarization of simulation software. Computing in Science & Engineering, 17(1), 38-45.

Bueler, E. (2014). An exact solution for a steady, flowline marine ice sheet. Journal of Glaciology, 60(224), 1117-1125.

Calvo, N., Díaz, J. I., Durany, J., Schiavi, E., & Vázquez, C. (2003). On a doubly nonlinear parabolic obstacle problem modelling ice sheet dynamics. SIAM Journal on applied mathematics, 63(2), 683-707.

Farrell, P. E., Kirby, R. C., & Marchena-Menendez, J. (2020). Irksome: Automating Runge–Kutta time-stepping for finite element methods. arXiv preprint arXiv:2006.16282.

Habermann, M., Maxwell, D., & Truffer, M. (2012). Reconstruction of basal properties in ice sheets using iterative inverse methods. Journal of Glaciology, 58(210), 795-808.

Jouvet, G., & Bueler, E. (2012). Steady, shallow ice sheets as obstacle problems: well-posedness and finite element approximation. SIAM Journal on Applied Mathematics, 72(4), 1292-1314.

Schoof, C., & Hewitt, I. (2013). Ice-sheet dynamics. Annual Review of Fluid Mechanics, 45, 217-239.

---

## Referee Report (RR1)

The new manuscript addresses all the points I was concerned with, and it is in very good
shape.  I look forward to trying-out Icepack with students.

Two small things.

lines 54-55:  Probable typo: "The evolution equations for ice thickness and velocity are
necessary for any simulation, BUT other fields with their own dynamics can be a part of the
problem as well."

lines 88-94:  Some thoughts, as follows.  It is not "erroneous" for an explicit prognostic
solver to compute negative thickness values, but rather inevitable, and even correct (as the
L^2 solution of the VI).  Also, the "principled" (i.e. implicit) approaches do not "track"
the free boundary, e.g. as an interface would be tracked in an immersed boundary method,
though the free boundary arises in the solution.  Thus the 6 sentences "A naively-
implemented ... through Firedrake." could be substantially simplified, e.g.:
"""
When our prognostic solver computes negative thickness values at a time step, where there is
ablation in ice-free regions, we truncate the thickness to zero, thus approximating the free
boundary.  An implicit approach could instead treat the free boundary problem directly as a
variational inequality (Jouvet and Bueler, 2012).  While Icepack currently lacks such a
scheme, it will be the subject of future development.
"""